# Prediction of future customer needs using machine learning across multiple product categories

David Kilroy[1]*, Graham Healy[2], Simon Caton[1]

1 School of Computer Science, University College Dublin, Dublin, Ireland, 2 School of Computing, Dublin City University, Dublin, Ireland

* david.kilroy1@ucdconnect.ie

## Abstract

In recent years, computational approaches for extracting customer needs from user generated content have been proposed. However, there is a lack of studies that focus on extracting unmet needs for future popular products. Therefore, this study presents a supervised keyphrase classification model which predicts needs that will become popular in real products in the marketplace. To do this, we utilize Trending Customer Needs (TCN)—a monthly dataset of trending keyphrase customer needs occurring in new products during 2011-2021 across multiple categories of Consumer Packaged Goods e.g. toothpaste, eyeliner, beer, etc. We are the first study to use this specific dataset and employ it by training a time series algorithm to learn the relationship between features we generate for each candidate keyphrase on Reddit to the ones in the dataset 1-3 years in the future. We show that our approach outperforms a baseline in the literature and through Multi-Task Learning can accurately predict needs for a category it wasn't trained on e.g. train on toothpaste, cereal, and beer products yet still predict for shampoo products. The findings from this research could provide many advantages to businesses such as gaining early access into markets.

**Data Availability Statement:** The dataset is accessible at https://github.com/davidkilroy/Multi-Task-Future-Customer-Needs-Model (page 3 of the manuscript). This includes links to freely access each resource detailed in the paper.

## 1 Introduction

Constantly driving innovation by producing new products is a critical success factor for Small and Medium-sized Enterprises (SMEs) [1] as well as large ones [2]. Businesses listen to the Voice of the Customer (VOC) to aid in the new product development/discovery process [3–5] as customer fulfillment is essential for the success of new products. Businesses are highly interested in identifying customer needs that are currently unmet [6] or anticipating/predicting future ones their customers may be unaware of [7]. Identifying these types of needs allows companies to gain early access into new markets and increase their overall profitability [8].

Over the last decade, researchers and businesses have been turning to the use of computational approaches to identify new customer needs in addition to traditional methods e.g. questionnaires, user observations, customer specifications, interviews, etc. [9]. Generally, these approaches mine User Generated Content (UGC) using statistical techniques from Artificial Intelligence (AI) and Machine Learning (ML) [10]. However, there is a lack of these techniques

**Funding:** DK, 18/CRT/6183, Science Foundation Ireland, https://www.sfi.ie/, No The funders had no role in study design, data collection and analysis, decision to publish, or preparation of the manuscript.

**Competing interests:** The authors have declared that no competing interests exist.

focusing on mining needs that are unmet or will be of importance in the future. Yet, approaches that can identify these types of needs are of major interest, for example, Black Swan Data (a firm specializing in predicting future market needs) accumulated over £15.2M in investments in 2022 alone—https://www.crunchbase.com/organization/black-swan-data/company_financials.

Computational approaches in the literature mainly mine sets of general customer needs which are posted as UGC [11–34]. However, these approaches fail to narrow their focus on identifying needs that are of substantial value to businesses e.g. unmet needs [16], needs of future importance [35], needs that can be turned into product opportunities [11, 13], etc. Unless the task is document classification, researched approaches typically don't employ supervised ML to extract customer needs instead opting to utilize unsupervised or rule-based approaches e.g. unsupervised clustering [11, 12, 20, 24, 36, 37] or rule-based keyphrase extraction [23, 38–41]. This is because supervised ML requires a ground truth dataset. Having this would most likely provide an increase in accuracy compared to previous approaches. Aside from the literature on customer needs mining needing a supervised approach, there seems to be a lopsided number of studies analyzing at the product model level rather than the product category level as pointed out in [35] (e.g. iPhone4 compared to mobile phones). This category-level analysis provides a different view as it allows needs in general products to be found rather than needs for a specific product model.

A major obstacle in attaining the mentioned aims is the lack of a supervised keyphrase classification model to predict customer future needs that are currently unmet. Therefore, in this study, we build a Multivariate Time Series Classification (MTSC) model which attempts to predict needs that will become popular in future products. To formulate this task, we utilize Trending Customer Needs (TCN) [42]—a dataset of trending keyphrase needs occurring in products each month from 2011–2021 which spans multiple product categories in the area of Consumer Packaged Goods (CPG) e.g. toothpaste, eyeliner, beer, etc. We are the first study to utilize this dataset and we use it by training a time series algorithm to learn the relationships between the keyphrases on Reddit to the ones in the dataset 1–3 years into the future. In our evaluation, we show that our approach outperforms a baseline from a previous study carrying out the same task. We also build a model that incorporates Multi-Task Learning (MTL) by being trained on multiple product categories (e.g. toothpaste, cereal, and beer) rather than just one category (e.g. toothpaste). This is significant as it can still predict accurately for a category it doesn't use during training e.g. can be trained on toothpaste, cereal, and beer yet still predict for cookies. By doing the following, our approach addresses the aforementioned limitations by predicting future customer needs occurring in multiple product categories using a supervised time series classification approach. There are 4 unique contributions of our work:

- The challenging task of predicting future customer needs is performed better than previous studies, allowing product development teams to identify unmet needs ahead of their competitors with greater accuracy.

- Due to the availability of the newly made TCN dataset, supervised ML is used to build a model capable of identifying future customer needs.

- The use of MTL is employed by incorporating data from multiple product categories in order to make predictions, which yields a model capable of predicting a category it doesn't use during training.

- Due to the availability of the TCN dataset our analysis spans many product categories. Consequently, it's also performed on the product category level (e.g. cheese) rather than the product model level (e.g. The Laughing Cow).

The remainder of the paper is organized as follows. Section 2 provides a literature review of the related approaches for extracting customer needs from UGC. Section 3 illustrates and describes our proposed method. Section 4 discusses the proposed evaluation and provides results. Finally, Section 5 concludes the study and discusses future research directions. We refer the reader to our GitHub repository for resources associated with the study which are mentioned throughout the paper—https://github.com/davidkilroy/Multi-Task-Future-Customer-Needs-Model.

## 2 Related work

Studies that mine customer needs from UGC use techniques from text mining, Natural Language Processing (NLP) and ML. These studies can be categorised based on: 1) Data Used; 2) Methods Performed; 3) Application Scenario; and 4) Evaluation. This section discusses studies in the area regarding the mentioned factors. Before discussing these techniques, we need to contextualise what is meant by a future customer need. Yet, we note that there is no universally accepted definition.

A customer need has been defined in the marketing literature as a "description in the customer's own words of the benefit to be fulfilled by the product or service" [43] e.g. the need to *prevent chapping* for Vaseline lip balm; i.e, they generally refer to requirements, demands, preferences, wants etc. Computational approaches analyzing UGC have also included the features or attributes of a product in this definition [44, 45] as they contain benefits e.g. the *coconut* flavour/scent which contains the need *fragrant* for Vaseline. In our study, the definition of a customer need is based on the output label from the TCN dataset [42] i.e. our ground truth label. In TCN, a customer need is in the form of a keyphrase, not a document or group of words/ phrases (i.e. topic) as in other studies. It groups needs into two categories: 1) direct needs— stated benefits as claims the user gets/overcomes from using the product (e.g. *prevent chapping*); and 2) indirect needs—actual features or attributes of a product which contain benefits (e.g. *coconut*). There are not many generic definitions in the literature of a future customer need. However, it is often mentioned alongside words like hidden [46] or unmet [47], hinting that they are sometimes undiscovered/unsolicited. In our study, we define it as a keyphrase that has direct/indirect benefits, but that this benefit will only obtained at some future time period. In our definition, this future time period is 1–3 years before the need starts trending in real products on the market (i.e. in the TCN dataset).

To clarify in our definition of a customer need, a keyphrase captures the main topics in a document [48–51]. It is different from a keyword as it connotes a multiword lexeme [51]. By extension, a candidate keyphrase is a phrase an algorithm analyzes to predict whether it is a keyphrase. In many computational studies, candidate keyphrases are initial sets of phrases that are first analyzed by an algorithm [48, 49, 51–53].

### 2.1 Data used

The main types of data used for extracting customer needs from UGC are social media [11–17, 54], product reviews [18–28, 55] and patents [29–34]. Social media has the drawback of containing a large number of posts that are irrelevant to customer needs when compared to product reviews and patents [35]. However, given the context of our research which aims to discover future needs, social media is the most suitable given that it's been used as a proving ground for new and emerging ideas e.g. social media users discussing do-it-yourself solutions to beauty products before they've become popularized in the market [56].

The social media platform Reddit is chosen for our analysis due to it being one of the few open opinion-based Application Programming Interface (API) platforms since the "Post-API

Age" [57] of platforms like Twitter and Facebook restricting access following major data scandals (e.g. Facebook Cambridge Analytica [58]) or changes in company policy (e.g. Twitter [59]). Specifically, when obtaining data we use the Pushshift API which has a limit "five times greater" [60] than the official Reddit API. We note that since collecting all the data needed for this article, API rules for Reddit have changed. At present it is unclear how Reddit data will be accessed in the future. Currently, it seems that a user licence (similar to that of Twitter) will be put in place for academics—https://techcrunch.com/2023/04/18/reddit-will-begin-charging-for-access-to-its-api. Baring data access, previous research using Reddit for customer needs mining [11–13] has noted the advantages of using the platform. It has stated the benefit of having data organized into defined "subreddits" when capturing needs where platforms like Facebook and Twitter are limited in this sense [11, 35] e.g. the subreddit r/PickAnAndroidForMe for Android products. In our study, we make use of this "subreddit" structure when classifying keyphrases as future customer needs. Documents are also generally longer on Reddit than on other platforms [11, 13] (e.g. Twitter's 280-character limit), which could potentially help in mitigating the short text problem [61, 62].

Many studies use ground truth data to train/evaluate ML algorithms capable of detecting customer needs from UGC. Depending on the task this type of ground truth data is different e.g. for document classification, a binary indicator may be included to indicate whether the instance contains a type of customer need [37, 63–70]. With our task being a keyphrase prediction problem, we instead require a dataset of ground truth keyphrases with an indicator of whether they are future customer needs.

To do this we use the newly available TCN dataset [42] which is specifically designed to train/evaluate models for discovering future customer need keyphrases. The dataset itself provides a set of the top 20 trending keyphrases each month from 2011–2021 for 37 product categories in the area of CPG e.g. toothpaste, eyeliner, beer, etc. (although we only use 15 categories in our analysis). It is constructed by having annotators label over 9000 keyphrases which are automatically extracted from a database of new-to-market product descriptions provided by Mintel Global New Products Database (GNPD) [71] (a large product information database used by industry and academics). By labelling needs from a dataset of product descriptions, TCN assures a certain quality of the keyphrases it provides as they are needs addressed in real products. The goal of this research is to predict the top needs occurring at a future time period in the dataset e.g. use Reddit data in 2015 to predict needs occurring in TCN in 2018.

The only other dataset that allows the training/evaluation of ML-based algorithms for predicting future customer needs in the form of keyphrases is [35], which also uses Mintel GNPD to annotate keyphrases for the evaluation of their approach. Unlike TCN, they annotate keyphrases using a named entity annotation approach, however, the output of the two datasets is the same which consists of a ranked list of trending keyphrases that are attempted to be identified ahead of time by algorithms run over UGC. One obvious drawback of the dataset in [35] is that it only is available for one product category (i.e. toothpaste) compared to TCN which evaluates for multiple categories. Due to this, TCN allows a major contribution of this research to be made possible which is in the use of MTL to build a single model that learns to predict future customer needs from any product category e.g. it can be trained on toothpaste, eyeliner and popcorn customer needs yet still predict toothpaste or even tea customer needs.

## 2.2 Methods performed

There are generally three families of methods used to mine for customer needs in the text mining and NLP literature: i) document classification; ii) clustering; and iii) keyphrase classification/ranking.

Document classification methods reduce the number of documents under analysis to ones that are "informative" from the standpoint of mining for customer needs. This definition of informative changes depending on the study at hand. In [63], this definition is based on whether the document contains a "wish" for making suggestions about an improvement for a product or an intention to purchase it. Similarly, studies on "purchasing intent" [64–67] identify documents showing "a desire to purchase a product or service in the future" [64]. Purchase intent studies mine on various UGC data sources such as Quora [64], Yahoo Answers [64] and Twitter [65, 66]. Other studies specifically classify documents based on whether they contain "customer needs" [37, 68, 69] from the definition in their respective studies. [68] gives examples of what customer needs are when obtaining labeled data for their classification task e.g. they state that the sentence "this product can make your teeth super-sensitive" is a need as it's informative from the sense of providing information about the product, however, "this product can be found at CVS" is uninformative as it only mentions the store it can be purchased in. Other studies have tried to classify documents that contain "product innovations" [70]. Methods used to solve these tasks range from classical ML methods (like Support Vector Machines, tree-based and Bayesian classifiers [69]) to deep learning approaches (such as Convolutional Neural Networks [68] or Long Short Term Memory Networks [70]).

Clustering methods can generally be split into three subfamilies of approaches: a) keyphrase clustering; b) document clustering; and c) topic modeling. Keyphrase methods group similar customer needs in the form of keyphrases together e.g. [36] clusters keyphrases in reviews from Amazon reviews of 4 smartphone products. Document methods group similar documents discussing customer needs together e.g. [20] clusters documents of reviews from Amazon for recliner products. Topic modeling can be seen as both keyphrase and document clustering as each document is a probability over topics, which are in turn distributions over keyphrases. They are used quite extensively in the literature, although used for various purposes e.g. analyzing fashion trends, smartphones or Amazon product ecosystems using Latent Dirichlet Allocation (LDA) [11, 12, 24, 37] or finding shorter-lived trends using LDA, Non-negative Matrix-Factorization (NMF), Latent Semantic Analysis (LSA) and neural topic models [72–74]. As our approach uses a keyphrase classification algorithm it doesn't necessitate using any clustering algorithms. However, it does borrow many techniques from these studies, for example, text preprocessing (e.g. Part-of-speech (POS) tag filtering [24]), document classification before running an ML algorithm (e.g. [37] ran an uninformative/informative review classification algorithm before running LDA), etc.

Approaches that work on the keyphrase level use various techniques and are applied for multiple purposes. A large body of work run over product reviews has focused on extracting a ranked list of the most important keyphrases [23, 38–41]. These keyphrases are found using rule-based approaches with various studies considering factors such as frequency or sentiment when ranking [23, 38]. However, the ranking of these keyphrases is not specifically designed for finding customer needs for product development, with many of these studies noting their uses for assistive purchasing information for future buyers based on previous ones [23, 38–41]. Other approaches do however focus on the ranking of keyphrases representing customer needs for product development [14, 15, 35, 75]. For example, [14, 15, 75] ranks and categorizes needs into strong, weak and controversial phrases from Twitter data for specific models of smartphone and automobile products (e.g. iPhone4, Motorola Droid RAZR, Tesla Model S, etc.). [35] builds on previous research by using a rule-based approach to compare their ranked list of customer need keyphrases extracted from Reddit to that of future needs extracted from a large database of toothpaste products. With the following studies using rule-based approaches to rank keyphrases from UGC data, one of the research gaps this study fills is in the usage of supervised ML to solve the ranking problem. We do this by leveraging the TCN dataset [42], a

recent benchmark that extracts top trending keyphrase needs between 2011–2021 from Mintel GNPD [71]—a database of new-to-market CPG products e.g. toothpaste, eyeliner, beer, etc. This dataset allows us to fit a supervised model to predict future trending keyphrases appearing in a set of new-to-market products from UGC. Although supervised ML has been used to classify documents in customer needs mining, it has not been used to predict trending keyphrases in this context before and is hence a contribution to our work. To classify keyphrases, we generate a group of time series representing features for each keyphrase e.g. an individual univariate time series representing some signal of sentiment, frequency, number of comments in the post, etc. We then classify these keyphrases using techniques from MTSC [76] based on whether they go on to trend in the TCN dataset at some future time period or not.

This study also explores the use of MTL, which has been described as having the aim of improving the learning of a model for a task by using the knowledge contained in a number of other learning tasks where these other tasks are related but not identical to the initial learning task [77]. The general approach of MTL has been applied in many applications of ML including but not limited to image classification [78], NLP tasks (such as sentiment analysis) [79, 80] and time series classification [81, 82]. It has also been used in unrelated customer needs mining tasks e.g. understanding customer needs from vehicle behaviour [83]. In our study we use it to learn an instance of a general customer need across multiple product categories, so that it can be used for a category it has or hasn't seen during training e.g. build a model on needs from toothpaste, lip balm and soda to predict for a seen category like toothpaste or even an unseen category like pizza. In our evaluation, we show how using MTL in this manner results in a high-performing model capable of accurately predicting categories it has and hasn't seen during training.

## 2.3 Application scenario

Other than methodological differences, studies in the area of customer needs mining also differ on the application level. Relevant to our study, is how other approaches differ on: i) the types of products analyzed; ii) whether the studies are based on previous methodologies in the business literature; and iii) the types of needs mined.

**2.3.1 Types of products analyzed.** Many studies in the literature tend to focus on analyzing customer needs for a specific product model such as smartphone products e.g. [11, 12] extract needs for the Samsung Galaxy Note 5 by mining on the subreddit r/galaxynote5. Other studies extract needs for multiple product models e.g. [14] finds needs for 4 smartphone models (e.g. iPhone4, Motorola Droid RAZR, etc.), [75] extracts for 4 automobile models (e.g. Tesla Model 5, Honda Civic, etc.) while [15] extracts for 4 models of smartphones and 4 models of automobiles. There is a lack of studies that analyze on the product category level however e.g. mine for general smartphone needs on social media rather than a particular model (e.g. iPhone4). Some that do include a document classification technique [68] which not only extracts needs on the category level but also for multiple categories i.e. toothpaste, kitchen appliances, skin treatment products and prepared foods. Likewise, [70] classifies documents containing innovation ideas across 20 categories of Amazon products. Similar to our approach, [35] extracts needs on the product category level using a keyphrase ranking approach. However, it only extracts and evaluates needs on one product category i.e. toothpaste. In contrast, our approach extracts and evaluates needs across 15 different product categories in the area of CPG. This is a required solution to show that a proposed approach generalizes beyond just one product category. To implement this training and evaluation across 15 product categories we make use of the aforementioned TCN dataset.

**2.3.2 Business methodology.** A major application scenario of customer needs mining has sought to provide automated solutions for previous models/methodologies defined in the business literature. In [11, 13], the idea of an "opportunity algorithm" is implemented to find new or existing customer needs. As initially described in [84], this algorithm works on the basis that if a need has high importance but low satisfaction then a business opportunity is present. In [11, 13], these importance and satisfaction values are computed based on the frequency at which a need is discussed (importance) along with the sentiment of it (satisfaction).

Kansei (Japanese for "affective") engineering is another business model that has been attempted to be automated by computational approaches. It deals with translating human emotions toward a product into design elements [85]. Previous business studies have implemented this method through the use of questionnaires, in which respondents rate their feelings on a point scale between two opposite pairs of words (one positive and one negative) known as Kansei attributes [86, 87]. However, recently automated studies using text mining and ML have tried to solve this problem using UGC to remove the time it takes to gather requirements by carrying out questionnaires [20, 88–90]. For example, [88] shows that their algorithm run over Amazon product reviews can extract emotions towards a customer need with high precision and recall.

Another business model addressed by computational methods is the Kano model [91], which is used in product development to weigh how much needs satisfy/dissatisfied customers. There have been many computational approaches to implement this model [18, 22, 37, 92–94]. For example, [37] applied sentiment analysis to the output returned by LDA to get the levels of satisfaction and dissatisfaction of customer needs in the form of topics. In our analysis, we don't implement any of the mentioned business methodologies specifically, however, we do use a particular study in Kansei engineering [88] to detect emotional words in UGC documents which contributes to us providing more features to our MTSC model for the prediction of future customer needs.

**2.3.3 Types of customer needs.** Many of the studies mining customer needs in the literature don't go beyond detecting general needs in UGC to identify ones that may be of more business interest e.g. specifically looking for unmet needs. For example, some document classification studies only detect "purchasing intent" [64–67] or only distinguish posts containing a customer need [68, 69, 95] without determining whether the document contains information that is of higher business interest e.g. contains an innovation that can disrupt the market. Similarly, some clustering approaches only detect groups of documents/terms forming needs discussed in UGC without highlighting ones that have more value e.g. [24] detects groups of general fashion needs in Amazon and Rakuten reviews. The same can be said for some keyphrase ranking approaches which consider the factors of frequency and sentiment when sorting phrases without specifically identifying ones that are of perceived business value [23, 38–41].

Some studies do however specifically focus on mining needs that are of business interest. [70] classifies documents that contain customer needs detailing product innovations which could be seen as more important than detecting general needs. In [11, 13], the aforementioned "opportunity algorithm" is implemented which identifies unmet needs by finding ones with high importance (keyphrase frequency) and low satisfaction (sentiment). This approach to identifying needs of greater interest to business builds on previous literature, however, it has been described as simplistic [84, 96, 97], and is criticized for this reason i.e. not all unmet needs conform to having high importance and low satisfaction. Similarly to the opportunity algorithm, the Kano model also goes beyond detecting general customer needs by providing a categorization/prioritization framework of needs into 3 groups (although some studies extend to more): 1) basic/must-have—needs, if left unfilled, will lead to dissatisfaction; 2) performance/one-dimensional—needs which give a proportionate increase in satisfaction as they are

invested in; and 3) excitement/attractive—needs which give no decrease in satisfaction if not fulfilled but may give disproportionately high satisfaction if fulfilled [98–100]. There are many computational approaches to implementing the classification of these need types using the Kano methodology from UGC [18, 22, 37, 92, 93]. For example, [92] classifies LDA topics into the 3 mentioned categories including 2 more (reverse and indifferent). The studies implementing this model go beyond just detecting general needs by providing businesses with more information on whether they should use them in their product and hence may be of more use in certain product development scenarios.

Comparable to clustering approaches, studies in keyphrase ranking also attempt to identify unmet needs, however, using different ideologies. In [35], unmet needs are detected using a rule-based approach by attempting to predict what needs will go on to be heavily addressed in products up to 3 years in the future. This idea goes on the basis that future needs are currently unmet by consumers and therefore finding them is of interest to businesses. Other approaches have used this principle but have applied regression techniques rather than keyphrase ranking to solve the problem [101–103]. Similarly, our approach uses the idea of attempting to find unmet needs by predicting ones that will be popular in the future. To do this, we use the TCN dataset which has been specifically designed for this purpose i.e. finding future customer needs.

## 2.4 Evaluation

In the customer needs mining literature, the difficulty of defining an evaluation strategy depends on the task being solved. For document classification tasks (e.g. does this document contain a "customer need"), the evaluation is straightforward where ML metrics such as accuracy, precision, recall and F1 can be employed [37, 63–70]. However, for tasks such as clustering or topic modeling the evaluation can be more difficult to define i.e. how do you say that a cluster/topic of keyphrases represents a customer need that is useful for product development? Due to this, some studies don't evaluate their approaches at all and instead demonstrate the usage of their approach [11–13]. Others use general intrinsic validation measures such as the Bayesian Information Criterion (BIC) [104] or the Silhouette score [17] to evaluate clustering solutions or perplexity to evaluate topic models [25, 37, 105, 106]. However, these studies don't show the validity of their solution in terms of evaluating extrinsically against clusters/topics of manually labeled customer needs, which is a necessary measure to employ in Information Retrieval (IR) tasks [107]. The lack of a benchmark dataset to evaluate clusters/topics has been a recently mentioned issue of studies evaluating these techniques [42].

When evaluating our approach the two main branches of literature we consider are those for 1) keyphrase-based evaluation; 2) future customer needs evaluation. This is done as our approach is a combination of the two of these areas in the general landscape of customer needs mining. One common methodology applied to evaluate keyphrase algorithms is an examination of the top set of keyphrases produced to obtain performance metrics [14, 15, 75] e.g. [14] examines a list of the top keyphrases generated from tweets. One of the main pitfalls of this approach is that it is only able to calculate precision but not recall, as a list of the total set of needs is not generated in the evaluation [42]. Other approaches do generate a list of ground truth keyphrases to calculate recall and precision [23, 38–41], but, they don't generate these lists with the specific task of mining needs for product development [42] (however they do help guide our evaluation). Approaches that predict future customer needs are mainly framed as a regression problem [101–103] which then go on to apply metrics such as Prediction Error [101] or Mean Absolute Percentage Error [102] to evaluate their approaches. These approaches have large prediction errors indicating the difficulty of the future customer need prediction

problem. As pointed out in [35], a problem with these approaches is that they don't predict a meaningful dependent/target variable for product development (e.g. sales), and instead forecast other variables based on the sentiment or frequency of a keyphrase in UGC.

To evaluate our approach, we use the TCN dataset—a set of the top 20 keyphrases representing customer needs each month from 2011–2021 across multiple product categories. This dataset allows for an evaluation approach that is keyphrase-based and that evaluates future customer needs. When using it for evaluation, we use traditional ML metrics (i.e. precision, recall, F1) in such a way that our algorithm can be assessed to identify future customer needs. In addition, we compute metrics that are used to evaluate lists of future customer needs. These metrics are the same ones calculated in the baseline approach [35], that we compare our algorithm to during our evaluation. The baseline is the only other available approach we are aware of which predicts future customer needs as keyphrases. The metrics calculated in the baseline are based on previous literature and represent a formulation of precision and recall that can be used for evaluating ranked lists i.e. List Mean Precision and List Recall. They are also the recommended metrics to use in the TCN dataset study [42], which advises a repeatable evaluation methodology to be used so that other researchers can benchmark their algorithms. The results in [35] show the difficulty of identifying future customer needs with the algorithm achieving a List Mean Precision result in the range of 10.3% to 15.8% and a List Recall result of 2.1% to 4.6%. These results may seem low, however, given the quality and difficulty of the evaluation approach (as discussed in [35, 42]), they are expected given the difficult task of identifying future customer needs. Having an algorithm with this performance may seem of low benefit at first, however, given the outcome of potentially identifying future customer needs, which can be highly profitable for businesses [7], its value becomes more apparent.

Table 1 summarizes the key studies mentioned in this section and shows the advantages of our approach and where it addresses the gap in the research when compared to other studies. The table shows that contributions are made in 1) finding unmet needs; 2) using supervised learning and MTL to mine for customer needs; 3) evaluating on the product category level (e.g. mobile phone) as opposed to the product model level (e.g. iPhone 2); and 4) evaluating using multiple product categories (e.g. lip balm, toothpaste and beer compared to just beer).

## 3 Methodology

Fig 1 outlines the keyphrase prediction problem addressed in this study. In this section, the main approach used to tackle this problem is described. In brief, the proposed approach aims to extract candidate keyphrases from social media data that predict keyphrases representing future customer needs i.e. in a future time period within TCN. The task itself is the exact one performed in [35] (i.e. future customer needs prediction), however, it's analyzed on several different product categories in our evaluation with the use of MTL, whereas [35] just evaluates it

**Table 1. Summary of studies in the customer needs mining literature.**

| Study | Technique | Data | Unmet Needs | Supervised ML | MTL | Category Level | Multiple Product Categories |
|---|---|---|---|---|---|---|---|
| [68] | Document Classification | Product Reviews (Amazon) | No | Yes | No | Yes | Yes |
| [11] | Topic Modeling | Social Media (Reddit) | Yes | No | No | No | No |
| [13] | Topic Modeling | Social Media (Reddit) | Yes | No | No | No | Yes |
| [23] | Keyphrase Ranking | Product Reviews (Multiple) | No | No | No | No | Yes |
| [15] | Keyphrase Ranking | Social Media (Twitter) | No | No | No | No | Yes |
| [35] | Keyphrase Ranking | Social Media (Reddit) | Yes | No | No | Yes | No |
| Our Approach | Keyphrase Ranking | Social Media (Reddit) | Yes | Yes | Yes | Yes | Yes |

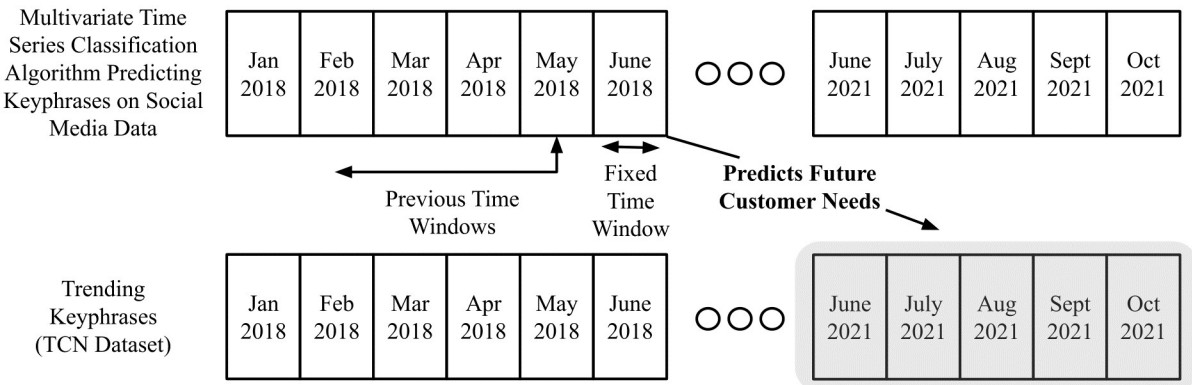

**Fig 1. Overview of task: Using past social media data to predict trending keyphrases in future customer needs addressed in products i.e. in the TCN dataset.**

on one i.e. toothpaste. The algorithm makes use of the timestamp associated with each social media post to make predictions at each Fixed Time Window i.e. predict keyphrases as customer needs each month (as in Fig 1). For the social media algorithm to make predictions, it considers data from Previous Time Windows. This is seen in Fig 1, where the algorithm uses data from 3 Previous Time Windows (i.e. April 2018, May 2018 and June 2018) to produce its final predictions for an individual Fixed Time Window (i.e. July 2018). Data from these Previous Time Windows is required due to the nature of the method being used (requiring past data for its computation). The overall prediction task is then to observe whether keyphrases from social media can be used to predict ones that appear in the TCN dataset at a future time period. This is seen in Fig 1 where the keyphrases predicted in the Fixed Time Window by the algorithm run over the social media data (i.e. June 2018) predict the keyphrases in the TCN dataset for a specific product category (i.e. June 2021, July 2021, August 2021, etc.). It is of note that the exact time frame seen in Fig 1 is not the one used in the experimental setup but is rather used to illustrate how the task is performed.

In our experiments, for each product category in the analysis, the TCN dataset is used to train and evaluate a supervised keyphrase extraction model run over social media to identify future customer needs. This is framed as a binary classification problem by checking whether a keyphrase on social media appears in the TCN dataset in some future time period for each product category i.e. keyphrase does/doesn't appear in the TCN dataset in the future. The significance of this is that the TCN dataset contains top keyphrases addressed in products and therefore by predicting what will occur in it we effectively forecast what new customer needs will be heavily addressed in future products e.g. predict the top needs for breakfast cereal items.

Fig 2 outlines the algorithm's approach when predicting future customer need keyphrases at each Fixed Time Window on social media for a specified product category. First social

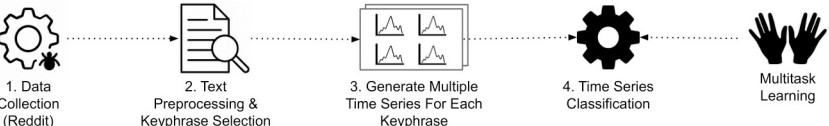

**Fig 2. Overview of methodology used to predict keyphrases representing future customer needs.**

media data is collected from Reddit (e.g. collect a corpus of posts for the category "soup"). Secondly, the posts are cleaned using text preprocessing techniques (e.g. tokenization, lemmatization, etc.) and candidate keyphrases are selected for the classification task. Thirdly, a large number of univariate time series are generated for each candidate keyphrase for the MTSC task. These series range from linguistic-based (e.g. dependency or part-of-speech tags) to sentiment-based information series. Finally, an MTSC model is trained on the univariate series associated with the candidate keyphrases which contain the binary output label determined by whether it appears in the TCN dataset in the future i.e. is a future customer need. This general framework of classifying keyphrases from social media using MTSC techniques has been addressed before with [108] generating similar families of features to our study (e.g. sentiment and content-based) to distinguish between "organic" and "promoted" hashtags on Twitter. A lot of the core concepts employed in the methodology of [108] are also used in our study, especially when generating univariate time series for each candidate keyphrase (or hashtag in their study). However, the types of univariate time series we generate in our study are tailored for the task of finding future customer needs e.g. time series encoding whether a keyphrase occurs in posts discussing products to help the model find keyphrases which are customer needs.

All of the numbered components in Fig 2 make up the subsections of this section with the addition of a further section to explain the types of univariate time series created to solve the classification task: 1) Data Collection—scraping Reddit data for each product category analyzed (Section 3.1); 2) Text Preprocessing & Keyphrase Selection—preprocessing posts and selecting candidate keyphrases (Section 3.2); 3) Generate Multiple Time Series For Each Keyphrase—discussing how time series are generated for each keyphrase for the classification task (Section 3.3); 4) Families of Univariate Time Series Generated—examining the families of univariate time series computed for the task of identifying future customer needs (Section 3.4); and 5) Time Series Classification—detailing how the ground truth label is added in the binary classification set-up and reviewing the MTSC algorithm applied (Section 3.5). As shown in Fig 2, we also discuss the use of MTL which is one of the main contributions of this study (detailed in Section 3.5). Here we describe the approach of building a single model capable of accurately predicting future customer needs in any product category.

## 3.1 Data collection

The social media data we use in our study is from Reddit. Some previous approaches using Reddit have looked at specific subreddits when mining customer needs for a specific product category [11–13] e.g. r/mobiles for mobile phone products. Instead, our approach searches for posts with a target keyphrase(s) that represents the product category in the analysis [16, 35] e.g. the target keyphrases "cookie" and "biscuit" are searched for when analyzing the Cookie product category. We confirm that we have permission to use all the data collected in this analysis. We also confirm that the collection and analysis methodology complies with the terms and conditions of the data owners. This work received an ethics waiver from the UCD Human Research Ethics Committee—Sciences (HREC-LS) under ref LS-E-20-81-Kilroy-Caton.

Table 2 shows the 15 product categories we analyze in this study. We collect data for each of these categories from 2011-01-01 to 2018-12-31. When choosing categories to analyze, we are restricted to ones that are in TCN (i.e. ground truth dataset)—initially totaling 37 CPG product categories. From these 37, we are further restricted due to some categories having too few Reddit posts to analyze them e.g. the product category Dishwashing Liquid has 14,161 posts from 2011-01-01 to 2018-12-31 with the searched target keyphrases: "dishwashing liquid", "washing up liquid", "wash up liquid", "dishwasher detergent" and "dishwashing detergent". From the remaining categories in TCN, we identified 15 of these for analysis to reduce

**Table 2. Overview of *Product Categories* used in analysis along with the corresponding: a) searched *Target Keyphrase(s)* on Reddit; b) total *Number of Posts* (rounded to nearest thousand) for each category we collect on Reddit; and c) the date the TCN ground truth is available.**

| Product Category | Reddit | | TCN |
|---|---|---|---|
| | *Target Keyphrases(s)* | *Number of Posts* | *Date Range* |
| Beer | beer | 480,000 | 2018-01-01 **to** 2021-12-31 |
| Cereal | cereal | 448,000 | 2018-01-01 **to** 2021-12-31 |
| Coffee | coffee | 480,000 | 2018-01-01 **to** 2021-12-31 |
| Cookie | cookie, biscuit | 471,000 | 2018-01-01 **to** 2021-12-31 |
| Dog Food | dog food | 159,000 | 2007-01-01 **to** 2021-12-31 |
| Eyeliner | eyeliner, eye liner | 347,000 | 2007-01-01 **to** 2021-12-31 |
| Lip Balm | lipbalm, lip balm, chapstick | 193,000 | 2007-01-01 **to** 2021-12-31 |
| Nail Polish | nail polish, nail varnish, nail lacquer | 248,000 | 2007-01-01 **to** 2021-12-31 |
| Perfume | perfume, fragrance | 360,000 | 2007-01-01 **to** 2021-12-31 |
| Pizza | pizza | 432,000 | 2018-01-01 **to** 2021-12-31 |
| Popcorn | popcorn | 416,000 | 2018-01-01 **to** 2021-12-31 |
| Shampoo | shampoo | 480,000 | 2007-01-01 **to** 2021-12-31 |
| Soda | soda, soft drink | 471,000 | 2018-01-01 **to** 2021-12-31 |
| Soup | soup | 465,000 | 2018-01-01 **to** 2021-12-31 |
| Toothpaste | toothpaste | 290,000 | 2007-01-01 **to** 2021-12-31 |

computational requirements. For the 15 categories, we select a diverse range of category classes ranging from the ones stated within TCN i.e. Health & Beauty (e.g. eyeliner), Pet (e.g. dog food), Food (e.g. cookie) and Drink (e.g. beer). This is done to show that our approach can work on multiple different classes of product categories e.g. not just Food and Drink.

Table 2 also shows the target keyphrase(s) searched for when collecting Reddit posts. When deciding which keyphrases are to be used as search terms we first include the product category name as a target keyphrase e.g. "beer" for the category Beer. Secondly, we include any obvious synonyms of the category name e.g. "soft drink" for the category Soda. Finally, we include any potential spelling variations or misspellings of the category name as it could account for a substantial number of posts missed on Reddit e.g. "eye liner" for the category Eyeliner.

Additionally, Table 2 shows the total number of posts we collect for each product category. To keep computational requirements reasonable for the following stages of the approach, we limit the number of posts to analyze for each product category. We do this by randomly sampling posts for each category for which we scrape data. Specifically, we employ disproportionate stratified random sampling at each Fixed Time Window (or strata) [109, 110]. That is to say, in our case, we sample a maximum number of posts at each Fixed Time Window regardless of its size proportional to the total number of posts. We do this to ensure that there is a sufficient number of posts in each Fixed Time Window. Specifically, the maximum sampling rate at each Fixed Time Window is 5000 posts, even if some of the windows don't have this amount of posts e.g. early in 2011 when Reddit uptake was low.

Furthermore, Table 2 shows the dates for which the ground truth data from TCN is available for each product category. Unfortunately, TCN only started collecting ground truth data for some categories on 2018-01-01. This makes some of the experiments in our study more challenging as it's not possible to use these categories in the current evaluation set-up of our model training process (detailed in Section 4). These categories can be used in the model testing process, however, hence their inclusion in the analysis.

We refer the reader to our GitHub repository (Section 1) for the release of Reddit IDs associated with each post from every category analyzed in the study.

## 3.2 Text preprocessing & keyphrase selection

For each product category analyzed, text preprocessing and keyphrase selection techniques are applied to automatically choose keyphrases from the social media post data—required to perform the future customer need classification task. For all the main preprocessing tasks implemented in this section, the Python library spaCy is used [111] i.e. sentence boundary detection, lemmatization and POS tagging. Specifically, the *en_core_web_lg* model from spaCy trained on OntoNotes 5 [112] is used to perform these tasks, which achieves high performance across many general NLP problems.

Fig 3 outlines the steps we perform to select candidate keyphrases from social media posts. The first step we carry out is sentence boundary detection [113] i.e. splitting a post into an array of sentences. We do this as we only extract the sentence where the searched "Target Keyphrase(s)" (i.e. Table 2) is mentioned e.g. when mining for the Lip Balm product category we only search for sentences containing "lipbalm", "lip balm" or "chapstick". As in [35], we do this as posts on Reddit can be quite large, with much of the discussion unrelated to the product category being analyzed.

Secondly, the sentence is tokenized, lemmatized and uncased, as performed in many other studies using keyphrase extraction [114–116]. Tokenization is needed as it is the first step required to separate candidate keyphrases for the classification task, while lemmatization and uncasing are carried out to group inflected phrases together.

Thirdly, multi-word phrases are formed by taking the set of all possible consecutive n-grams in the range of 1 to 4 grams [117, 118]. This process is seen in Fig 3 by forming multi-word phrases from unigrams e.g. "coconut_lip" from the consecutive words of "coconut" and "lip".

Finally, only n-grams with specific POS tag combinations are kept for the next stages of the analysis, as in [119, 120]. For our task, we require tag combinations that correspond to customer needs. To do this, we extract tag combinations of phrases that are already labelled as customer needs in the TCN dataset. Specifically, we extract all the keyphrases recorded across 5 product categories in the dataset i.e. Vitamins & Dietary Supplements, Cat Food, Pasta Sauce, Tea and Potato Snacks. These categories are not used in the primary analysis (Table 2)

**1. Sentence Split**

It was a wonderful walk. The coconut lip balm prevented my lips from being chapped. Luckily it started raining as soon as we got to the car.

**2. Tokenization, Lemmatization & Uncasing**

["the", "coconut", "lip", "balm", "prevent", "my", "lip", "from", "be", "chap", "."]

**3. Ngrams**

["the", "coconut", "lip", "balm", "prevent", "my", "lip", "from", "be", "chap", ".", "the_coconut", "coconut_lip", … "lip_from_be_chap", "from_be_chap_."]

**4. POS Filtering**

["coconut", "lip", "balm", "prevent", "lip", "chap", "coconut_lip", … "coconut_lip_balm_prevent"]

**Fig 3. Overview of text preprocessing & keyphrase selection in order to extract candidate keyphrases.**

to avoid any potential bias in our experimental evaluation. They are also diverse in category classes including Pet Food (i.e. Cat Food), Drink (i.e. Tea), etc. This diversity is necessary as POS tag combinations associated with keyphrase customer needs are different across category class types e.g. the POS tag combinations in Pet Food are different to Drink tag combinations. To extract these tag combinations from the keyphrases in TCN, we run the same *en_core_web_lg* model over them to identify their POS. In total, we identified 31 tag combinations from the 5 product categories. These are made up of single-word combinations (e.g. nouns like chicken or adjectives like energetic) as well as multi-word combinations (e.g. adjective phrases like micro-cleaning). All the POS tag combinations identified are contained within the single POS tags of nouns, verbs, adjectives, adverbs or proper nouns. This is expected as the TCN dataset instructed annotators to only label customer needs with these POS tags [42] —https://github.com/davidkilroy/TCN-Dataset. A complete list of these POS tags can be found in the GitHub repository which accompanies this study (Section 1). It's important to note that there is similar work for generating task-specific POS tag combinations for keyphrase extraction e.g. [119] generates a list of tags for the extraction of computational linguistic terms.

### 3.3 Generating multiple time series for each keyphrase

In this section, we describe how we transform the collection of preprocessed posts for each product category (discussed in Section 3.2) to a form suitable for the prediction of keyphrases using techniques from MTSC. Fig 4 shows an example output of the data we produce. As with classical ML, we generate several features for each candidate keyphrase. However, for our task, the value in the fields generated for each feature is not an individual number but a univariate time series. For each candidate keyphrase instance, each of these univariate series makes a multivariate series (required for the task of predicting future customer needs using MTSC techniques). In this section, we solely describe the process of going from a collection of preprocessed posts (i.e. output of Section 3.2) to multivariate time series data for each keyphrase (in Fig 4). In the next sections, we discuss the types of time series features generated (Section 3.4) before detailing how the ground truth label from the TCN dataset is added to each candidate keyphrase instance along with the MTSC techniques used to classify them (Section 3.5).

Fig 5 shows a top-down view of the transformations performed to move from a collection of preprocessed posts to a set of candidate keyphrases with multiple associated features in the form of univariate time series. Each of the steps in the figure will be described in this section. At a high level we do the following: 1) Add Additional Features—text-based models are run over the post data e.g. running a text classification "Buy Intent" model from HuggingFace over posts; 2) Group Keyphrases & Summarize Features—features are summarized for each

| Keyphrase | Feature 1 | Feature 2 | ... | Feature 1000 |
|---|---|---|---|---|
| charcoal | [0.02 . . . 0.05] | [0.6 ... 0.8] | ... | [1 . . . 1] |
| bread | [0.02 . . . 0.01] | [0.3 . . . 0.2] | ... | [0.85 . . . 0.9] |
| milk | [0.05 . . . 0.06] | [0.7 ... 0.9] | ... | [1 . . . 0.95] |
| orange | [0.001 . . . 0.0002 | [0.3 ... 0.25] | ... | [0.9 . . . 0.95] |

**Fig 4. Example output of data suitable for multivariate time series classification.**

| Date | Post | Post Preprocessed | Post Type (sub/com) | Score Attribute |
|---|---|---|---|---|
| 2011-01-29 | "...my homemade charcoal toothpaste.." | ["homemade", "charcoal", "toothpaste" ... "charcoal_ toothpaste"] | sub | 5 |
| 2015-08-14 | "...coconut toothpaste from.." | ["coconut", "toothpaste", "coconut_ toothpaste"] | com | -4 |

Step 1: Add Additional Features

| Date | Post | Post Preprocessed | Post Type (sub/com) | Score Attribute | Buy Intent - Hug Face | Doc Embed Dim 1 - sbert | Doc Len - spaCy |
|---|---|---|---|---|---|---|---|
| 2011-01-29 | "...my homemade charcoal toothpaste.." | ["homemade", "charcoal", "toothpaste" ... "charcoal_ toothpaste"] | sub | 5 | 0.8 | 0.45455 | 25 |
| 2015-08-14 | "...coconut toothpaste from.." | ["coconut", "toothpaste", "coconut_ toothpaste"] | com | -4 | 0.8 | 0.2342 | 50 |

Step 2: Group Keyphrases & Summarize Features

| Date | Keyphrase | Doc Freq | Sub % | Mean Buy Intent | Max Buy Intent | Min Buy Intent | Sum Buy Intent |
|---|---|---|---|---|---|---|---|
| 2011-01-01 | charcoal | 0.02 | 0.6 | 0.7 | 1 | 0 | 10 |
| 2011-01-01 | bread | 0.02 | 0.3 | 0.2 | 0.85 | 0 | 2 |
| 2018-12-01 | charcoal | 0.05 | 0.7 | 0.08 | 1 | 0 | 20 |
| 2018-12-01 | bread | 0.001 | 0.3 | 0.05 | 0.9 | 0 | 5 |

Step 3: Turn Into Time Series

| Date | Keyphrase | Doc Freq | Sub % | Mean Buy Intent | Max Buy Intent | Trend in TCN 1-3 Years in the Future? |
|---|---|---|---|---|---|---|
| 2014-01-01 | charcoal | [0.02 ... 0.05] | [0.6 ... 0.8] | [0.7 ... 0.8] | [1 ... 1] | Yes |
| 2014-01-01 | bread | [0.02 ... 0.01] | [0.3 ... 0.2] | [0.2 ... 0.1] | [0.85 ... 0.9] | No |
| 2018-12-01 | charcoal | [0.05 ... 0.06] | [0.7 ... 0.9] | [0.08 ... 0.75] | [1 ... 0.95] | Yes |
| 2018-12-01 | bread | [0.001 ... 0.0002] | [0.3 ... 0.25] | [0.05 ... 0.03] | [0.9 ... 0.95] | No |

**Fig 5. Overview of preprocessing to extract candidate keyphrases.**

candidate keyphrase at each Fixed Time Window (i.e. month) e.g. for the keyphrase charcoal calculate the mean "Buy Intent"; and 3) Turn Into Time Series—for each keyphrase at a given Fixed Time Window (i.e. month) the features are turned into individual univariate time series by obtaining the values each month for the keyphrase of interest 36 months into the past (i.e. number of Previous Time Windows) and sorting them by time e.g. for charcoal on the 2014-01-01 find the "Mean Purchase Intent" each month from 2011-01-01 to 2014-01-01. As previously discussed, this entire generation process is performed the same way as [108] by

constructing time series for keyphrases (or hashtags in their case) based on calculating summary statistics from the posts it occurs in at each Fixed Time Window. We do, however, generate different time series in our study to reflect the task of recognizing future customer needs (discussed in Section 3.4), which is different from classifying between "organic" and "promoted" hashtags [108].

The first step of the approach is to generate several additional features about the post or candidate keyphrases of interest in the post. When calculating post-level features, we apply various text-based models to the sentence with the target keyphrase and record its output. As seen in Fig 5, the models we choose may record document-level information such as Buy Intent from the library Hugging Face—distinguishing between posts containing "buy intent". We include these features as we believe there is a justification for them improving the task of predicting future customer needs. A detailed list of all the features used along with reasons why they are added is addressed in the next section (Section 3.4).

The second step is to group candidate keyphrases in the posts together by Fixed Time Window (i.e. month) and calculate summary statistics for them based on the posts they appear in. This is seen in Fig 5, where we compute the "Mean Buy Intent" for the keyphrase charcoal on 2011-01-01. Depending on the data type of the feature being summarized different summary statistics are computed. For example in Fig 5, the mean can be calculated for the "Buy Intent" feature as it's a continuous feature, however, it can't be calculated for the "Post Type (sub/com)" feature as it's a string. Here "sub" and "com" refer to Reddit submissions (main posts) and comments (comments on submissions).

In total, we calculate summary statistics for four different types of features (three of which are data types): 1) continuous features, 2) boolean features, 3) string features and 4) keyphrase-level features. Continuous features are columns where the fields contain numerical values (e.g. 4, 5.1, etc). For these features we compute the following summary statistics a) mean, b) maximum, c) minimum and d) sum. Boolean features are columns where the fields contain True or False values. For these features, we only compute the percentage of posts that are True as a summary statistic of the feature column. We only record True, as False can be inferred from it, and hence does not add any additional information to the ML model. In some situations, however, we record False along with the value Not a Number (NaN) as it appears in some fields provided by the Reddit Pushshift API (e.g. "is_video" field). String features are columns where the fields contain a sequence of characters e.g. "submission" for the "Post type (sub/com)" in Fig 5. For string features, we carry out type matching of a user-defined string and report the percent of posts that contain the matched string as a summary statistic e.g. percent of posts that are "submissions" based on the "Post type (sub/com)" feature (as in Fig 5). The process for choosing the types of strings to search for is dependent on the feature summarized. In some cases, it's an exhaustive list of all possible strings in the string-based feature column (e.g. "submission" and "comment" for the "Post type (sub/com)" feature) and in other cases, only specific strings are searched for due to there being too many values in the feature (e.g. subreddit feature column). Keyphrase-level statistics are obtained by retrieving an attribute from the keyphrase, therefore accounting for just one summarized statistic. As seen in Fig 5, an example of a keyphrase-level feature is the Document Frequency at a given Fixed Time Window. Another example may include a dimension of a word embedding for a keyphrase. Table 3 outlines the discussed summary statistics we compute in this study. It's important to note that it's possible to include additional statistics (e.g. median for continuous types), however, for computational reasons we decided to keep this number low. The current statistics are just put in place to test whether the general framework works i.e. to see whether it can learn what future customer need trends look like.

**Table 3. Overview of summary statistics used.**

| Type of Feature | Summary Statistic(s) | Num Series |
|---|---|---|
| continuous | 1) mean<br>2) maximum<br>3) minimum<br>4) sum | 4 |
| boolean | 1) % True<br>2) % False (optional)<br>3) % Not a Number (optional) | 2 or 1 |
| string | % Searched String | 1 |
| keyphrase-level | Keyphrase-level Statistic | 1 |

The third and final step of the approach is to turn each summary statistic feature into an individual univariate time series. This is done by obtaining the values each month for the candidate keyphrase of interest 36 months into the past (i.e. number of Previous Time Windows) for a specific feature and ordering these values by time e.g. charcoal on the 2014-01-01 find the "Mean Purchase Intent" each month from 2011-01-01 to 2013-12-31. As seen in Fig 5, the final output of this is multiple univariate time series for each candidate keyphrase instance per month (i.e. Fixed Time Window).

As we build time series based on data 36 months into the past (i.e. 36 Previous Time Windows), we reduce the time frame in which we have multivariate time series data. When collecting Reddit post data, we scraped between 2011-01-01 to 2018-12-31 for each product category. However, due to this construction step, we only have time series data from 2014-01-01 to 2018-12-31. Another consequence of the construction step is that instances with the same keyphrase for the same product category between Fixed Time Windows (i.e. months) are highly similar. This is a result of the rolling window nature of our approach when constructing the time series e.g. the two instances of "charcoal" in the product category Toothpaste for the months 2014-05-01 and 2014-06-01 are nearly the same. As we described later in our evaluation (i.e. Section 4), due to this we are constrained to having the training and testing data separated by at least 36 Previous Time Windows for each instance i.e. the time period required for the instances to no longer share any time series data. Therefore in our evaluation, the training period for each category is between 2014-01-01 to 2014-12-31 and the testing period is between 2018-01-01 to 2018-12-31. We do this as 2014-12-31 and 2018-01-01 are separated by 36 months (avoids any potential train/test overlap that would occur). Because the training period is between 2014-01-01 to 2014-12-31, we can only train on 7 product categories in our analysis as there is only TCN ground truth available for these 7 for those dates (as seen in Table 2). Although the remaining 8 categories are not used in the training process they are used in our evaluation (Section 4).

Each month when generating candidate keyphrases, the approach also only considers keyphrases for the classification task that exceed a minimum frequency over the past 36 months (i.e. length of Previous Time Windows). Specifically, the minimum thresholds we apply are a minimum document frequency of 0.00005 (as in [35]) and a minimum raw count of 2 over the past 36 months. These thresholds are very lenient for a candidate keyphrase to pass as ones that don't pass these thresholds are unlikely to be future customer needs which trend in the TCN dataset i.e. are addressed as top customer needs in future products. By applying these thresholds it also allows us to summarize fewer candidate keyphrases (we cut off the long tail of the distribution), which leads to less computational resources being used.

So to obtain a better understanding of the process described in this section, we refer the reader to our GitHub repository (Section 1) for a small release of a random sample of our data

generated at each of the detailed steps in this section i.e. release of data at each of the processes in Fig 5.

## 3.4 Families of univariate time series generated

In our analysis, we can split the total number of unique univariate time series into families of series, as shown in Table 4. In total, there are 1263 univariate time series from 10 families of series. The idea behind including this large feature set is to learn what a future customer need instance looks like on the social media platform Reddit. In this section, we give an overview of these families of features with a rationale behind the selection process for their inclusion. The appendices give a more in-depth description of how these features are generated.

The series from the **Reddit Information Based Series** are generated from attributes that are provided with each post from Pushshift i.e. historical Reddit API [60]—https://pushshift. io/api-parameters/. These collected attributes range from the score (i.e. number of upvotes minus number of downvotes on a post) to whether the post contains a video. The majority of the time series generated here (e.g. a time series generated from an attribute about a post containing a video) may not necessarily be directly useful in the multivariate problem of detecting whether a keyphrase will become a future customer need addressed in real products. However, some features are directly useful, such as series derived from the score or the number of comments, with other studies on Twitter using retweet and like attributes to identify future product trends [121]. Refer to S1 Appendix for a more detailed description of the exact features created for the Reddit information-based series.

The time series from the **Frequency Based Series** are generated from different statistics about each candidate keyphrase's occurrence in each Fixed Time Window (i.e. month). All of these types of features are keyphrase level statistics (as described in Section 3.3) e.g. document frequency. Measures of keyphrase frequency have been used in previous tasks identifying customer needs from social media [35, 44] and therefore are useful in this classification task. Refer to S2 Appendix for a more detailed description of the exact features generated for the frequency-based series.

For the **Product Information Based Series** we generate features from pre-trained models which are run over Reddit posts. These models all try to capture whether a post is "product-related" in some sense e.g. post contains purchase intent [67]—https://huggingface.co/j-hartmann/purchase-intention-english-roberta-large. These types of features are good at

**Table 4. Overview of features used.**

| Family | Appendix | Number of Time Series |
|---|---|---|
| Reddit Information Based | A | 51 |
| Frequency Based | B | 4 |
| Product Information Based | C | 24 |
| Sentiment Based | D | 112 |
| Question Detection Based | E | 20 |
| Embedding Based | F | 350 |
| Subreddit Based | G | 100 |
| Kansei Engineering | H | 32 |
| Linguistic Based | I | 456 |
| User Based | J | 114 |
| | | **1263** |

identifying customer needs [68], hence their inclusion. Refer to S3 Appendix for a more detailed description of the exact features generated for the product information-based series.

For the **Sentiment Based Features**, as with Product Information Based Features, we generate features from a pre-trained model that is run over Reddit posts. Specifically, we summarize the outputs of a model run over the GoEmotions dataset [122], which contains 28 output class labels representing emotions (e.g. anger, caring, disappointment, excitement, etc.). Sentiment has been widely used in the customer needs mining literature [37, 105], hence its inclusion as a feature. Refer to S4 Appendix for a more detailed description of the exact features generated for the sentiment information-based series.

For the **Question Detection Based Series**, as with some other features discussed in this section, we generate features from models that are run over Reddit posts. These models try to detect whether a post is asking a question or stating an answer. These features are included with the hypothesis of future customer need keyphrases being in more posts that contain questions or statements e.g. people asking what charcoal toothpaste was before it became a popular customer need in toothpaste products—https://www.reddit.com/r/NaturalBeauty/comments/2s6h2u/best_homemade_whitening_toothpaste/. Refer to S5 Appendix for a more detailed description of the exact features generated for the question detection-based series.

For the **Embedding Based Series**, as with some other features discussed in this section, we generate features from models that are run over Reddit posts. We do this by using pretrained document and phrase embedding models with the Python libraries SBERT [123], spaCy [111] and fastText [124]. Embedding information has already been used to identify customer needs in other studies [68, 125] (although used for document classification). It's also feasible to say that it will provide predictive information for our classification task as past trending phrases likely share a similar vector space by having similar meaning (i.e. phrase embeddings) while customer need keyphrases may be in similar documents to past trending ones (i.e. document embeddings). Refer to S6 Appendix for a more detailed description of the exact features generated for the embedding-information based series.

For the **Subreddit Based Series**, we summarize subreddit (discussion forum on Reddit) information associated with each post. As the subreddit feature on Reddit is a string (e.g. r/AskReddit, r/Music), we search for defined strings to generate a statistic for each candidate keyphrase (as described in Section 3.3). The defined strings we search for come from 100 of the most subscribed subreddits at the time of experimentation, resulting in 100 new univariate time series features in the classification problem. The use of subreddit information has been applied in previous research using Reddit to identify future customer needs [35]. It is useful as certain subreddits may be indicative of places where new trends are discussed e.g. in the subreddit r/eli5 people may ask questions about queries they want solved such as best ingredients to use for smoother lips (lip balm) or whiter teeth (toothpaste). Refer to S7 Appendix for a more detailed description of the exact features generated for the subreddit information-based series.

For the **Kansei Engineering Based Series**, we classify posts based on whether they contain one of the words in a Kansei group. Kansei engineering has been described as "translating technology of a consumer's feeling and image for a product into design elements" [85]. Recently, it has become an important topic in the customer needs mining literature for product development, with many computational approaches to it being built [19, 20, 88–90]. Traditional non-computational approaches to Kansei engineering work on questionnaires to measure a user's feelings towards a customer need where groups of words called Kansei attributes are used to measure their emotions. Kansei attributes consist of a pair/groups of bipolar words in which respondents choose to indicate their feeling towards a product e.g. 1) unique-personalized-rare vs common-general; 2) quality-reliable-sturdy-safe vs unreliable or 3) novel-

fresh-interesting vs boring. We classify posts based on whether they contain one of the words in a Kansei group. To retrieve a list of these Kansei attributes, we follow the work in [88] which first identifies 16 groups of bipolar Kansei attributes from 10 previous Kansei engineering studies (mostly in the last decade) and then expands on these attributes using an automated method. Refer to S8 Appendix for a more detailed description of the exact features generated for the Kansei Engineering information-based series.

For the **Linguistic Based Series**, as with some other features discussed in this section, we summarize features from models that are run over Reddit posts. We also analyze keyphrase-level statistics. All the univariate series we generate either represent 1) tagging information (e.g. POS, dependency labels, etc.), 2) document information (e.g. post length) or 3) phrase-level information (e.g. the number of vowels, whether it contains an @ symbol, etc.). Refer to S9 Appendix for a more detailed description of the exact features generated for the linguistic information-based series.

For **User Based Series** we generate features based on authors (i.e. users on Reddit). The use of author information has been seen in many of the social media forecasting topics already discussed in this study e.g. predicting customer needs [95] and detecting future occurrences using MTSC [108]. Refer to S10 Appendix for a more detailed description of the exact features generated for the user information-based series.

### 3.5 Time series classification

In this section, we describe the time series techniques used to address the future customer needs keyphrase classification problem. Specifically, we discuss 1) how the ground truth label is added to each candidate keyphrase from the TCN dataset; 2) the MTSC algorithm used for the task (i.e. Supervised ML); and 3) the use of MTL to build a single model capable of identifying future customer needs in any product category.

As previously discussed, the TCN dataset consists of the top 20 most addressed customer needs in products each month from 2014-01-01 across multiple product categories. Fig 6 shows how we add this data as the ground truth label for each multivariate time series instance, where each instance is a candidate keyphrase in a given Fixed Time Window (i.e. month) for a particular product category (i.e. Toothpaste in Fig 6). A binary output label indicates whether the phrase appears in the TCN dataset for the product category being analyzed 1–3 years in the future. This is seen in the figure for the term "charcoal" on 2014-01-01 which has a positive output label (i.e. Yes) given that it appears as a top customer need in TCN3 years in the future i.e. in 2018-01-01. The next instance "bread" has a negative output label (i.e. No) as it doesn't appear in the TCN dataset during that time period. It's important to understand that the main objective of adding the ground truth label to the instances is to train and evaluate a MTSC algorithm that predicts customer needs ahead of time before they hit the marketplace (specifically 1–3 years ahead). The main premise behind this is that future customer needs in a

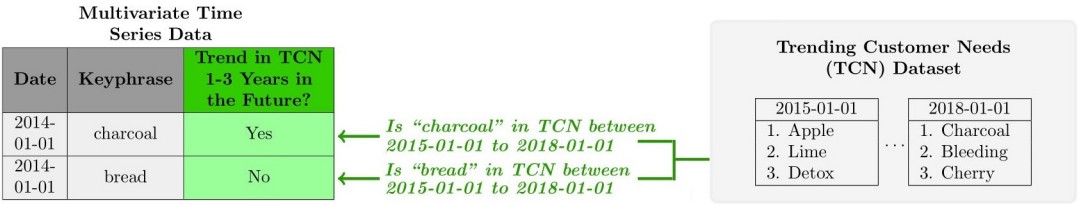

**Fig 6. How ground truth label is added for the classification problem from TCN.**

product dataset (i.e. TCN) represent needs that are currently unmet and are therefore valuable for businesses to identify. Although not seen in Fig 6, it's also valuable to point out that there is a high degree of data imbalance between the positive and negative label in the final ground truth column (i.e. "Trend in TCN1-3 Years in the Future?" column in Fig 6). To clarify, there are a lot more negative instances than positive ones because multiple thousand instances are being analyzed in each Fixed Time Window (i.e. month) and there are only 20 customer needs 1–3 years ahead each month in the TCN dataset. In Section 4, we discuss the techniques used to handle this data imbalance.

Unlike univariate time series classification where an instance is a time series with a number of temporally ordered observations and output class, multivariate time series classification is a list of vectors with a number of dimensions along with a number of observations and an output class [76]. As of 2018, an archive of 30 multivariate time series datasets has been released (diverse in series length, number of dimensions and number of output classes) allowing for the benchmarking of algorithms run over these data types which has led to an increase in research in this area [126]. Families of these algorithms have not been applied often when mining for customer needs, however, they have been applied in related areas of study e.g. smart manufacturing [127] and customer churn prediction [128]. Additionally, the inclusion of libraries implementing many popular algorithms in the field has been made publicly available, allowing studies showing the applicability of these algorithms to be made. Two popular ones include 1) sktime—a Python-based package compatible with sklearn [129] and 2) tsml—a Java-based package compatible with Weka. In our study, we use the multivariate supervised MINImally RandOm Convolutional KErnel Transform (MINIROCKET) algorithm [130], a faster version of the RandOm Convolutional KErnel Transform (ROCKET) algorithm [131], which has been shown to obtain better results in terms of speed and accuracy than comparative approaches [76]. ROCKET is an algorithm for transforming a 3D multivariate time series into a 2D vector space using random convolutional kernels. This 2D vector space is then used as ML features to train a linear classifier such as Ridge/Logistic Regression [131] to solve the classification task. In our analysis, we use MINIROCKET, which is a deterministic algorithm that speeds this ROCKET transformation process up to 75 times faster on large datasets [130]. When applying MINIROCKET, we use the multivariate version from sktime—http://www. sktime.net/en/v0.13.0/api_reference/auto_generated/sktime.transformations.panel.rocket. MiniRocketMultivariate.html. To train the linear classifier on the embeddings produced by MINIROCKET, the cross-validated version of Ridge Regression from sklearn is used (one of the recommended algorithms to use with MINIROCKET [130])—https://scikit-learn.org/1.0/ modules/generated/sklearn.linear_model.RidgeClassifierCV.html. We apply the following implementations of models from the mentioned libraries as they are the recommended ones used in the linked coding repository of MINIROCKET- https://github.com/angus924/ minirocket. We also use the same default hyper-parameter values for the two models as in the repository. Two of these important hyper-parameter values include: 1) 10,000 for the *num_kernels* parameter of MINIROCKET- producing an embedding space of 10,000 dimensions which the linear model is trained on; and 2) True for the *normalize* parameter of Ridge Regression—standardizes the embeddings before training/testing the classifier. It's important to note that although we normalise the embedding inputs into the linear classifier we do not normalise the multivariate time series data before running MINIROCKET (as performed in major MTSC benchmarking studies [76]). This is because scale and variance in one dimension within multivariate data may be discriminatory factors, which is particularly relevant to MTSC where interactions in shape, level and variance are required [76].

As discussed in Section 2, the use of MTL is a key contribution to our study. How we use it is described in Fig 7. During training, we generate time series features from the instances of

the available training product categories we have at our disposal (e.g. Dog Food, Shampoo and Toothpaste in Fig 7). A model is then built from these instances which contains the ground truth label which allows the prediction of future customer needs. During testing, we generate time series features using the same process during training, however only for one category. The described trained model is then used to classify these instances. We use two types of product categories when testing our model during evaluation: 1) *Seen Testing Category*—a category the model has seen during training (e.g. Dog Food in Fig 7); and 2) *Unseen Testing Category*—a category the model has not seen during training (e.g. Cookies in Fig 7). For the Seen Testing Category, although the model has used the category in the training process the same data is not used for training and testing—described at more detail in our evaluation (Section 4). As discussed in Section 3.3, the categories in Table 2 which have ground truth data on/before 2014-01-01 are used to train the model (i.e. 7 categories) and thus also make up the Seen Testing Categories. This is due to the fact the training time period in our evaluation is between 2014-01-01 to 2014-12-31 (for the reasons described in Section 3.3). All other categories are not used in model training and therefore make up the Unseen Testing Categories (i.e. 8 categories). In our evaluation, we show that the model which is produced from this described MTL approach leads to similar performance compared to training and testing on the same product category e.g. train on Dog Food to predict Dog Food. This is important as this model can be used on categories the model has not seen during training. It does this by learning what general future customer needs look like on Reddit rather than one for a particular product category. The reason this model performs better is due to the MTL characteristic of *Task Relatedness* [77] i.e. tasks are similar. In our task, this characteristic is seen due to the signals of future customer needs on Reddit for different categories being similar e.g. Toothpaste and Cookies. This is also the logic behind most of the MTL approaches working better throughout the ML literature e.g. [132] made a better-performing classification model which learned higher-level features by using MTL to train on images from multiple object categories. This characteristic of *Task Relatedness* for our problem is helped by how we generate task-agnostic features. This is seen in Section 3.4 where the features we generate are not specific to any one product category but rather general to multiple product categories e.g. user/frequency/sentiment features.

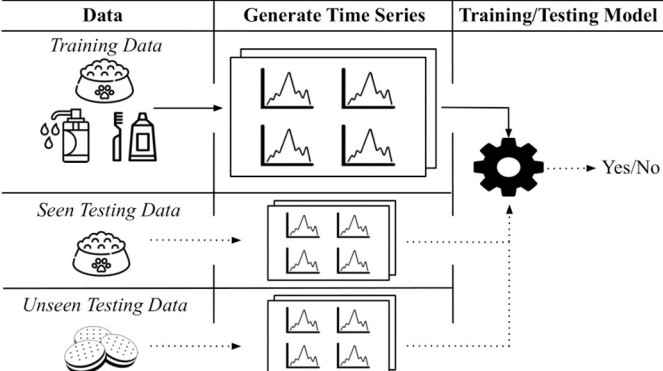

**Fig 7. Multi-Task Learning: A generalizable model is built from multiple product categories (e.g. Dog Food, Shampoo and Toothpaste) and tested on categories it has seen (e.g. Dog Food) and not seen during training (e.g. Cookies).**

## 4 Evaluation

This section aims to answer the following two research questions: 1) can future customer needs be predicted with better performance than previous approaches using UGC from the social media platform Reddit across multiple product categories within CPG?; and 2) can the use of MTL (described in Section 3.5) be employed to achieve similar performance to training/testing on the same category so that it can be applied to categories the model hasn't seen during training i.e. by learning what a general future customer need looks like across multiple product categories? To do this, we first measure our approach against a baseline in the literature [35] by assessing how the model compares to it when trained and tested on the same product category e.g. train and test on toothpaste. We then show the performance of MTL when compared to the approach of training and testing on the same category.

This section first details the two different training strategies we use in our experiments i.e. training on one versus multiple categories for prediction (Section 4.1). When describing the strategies, we also give an in-depth explanation of the specific model training and validation details used in our experiments. We then describe the evaluation approaches we employ and explain why certain metrics are used to assess performance (Section 4.2). Our approach is then compared to a baseline [35] in the literature which carries out the same future prediction task as in our study (Section 4.3). The impact of MTL is then assessed (Section 4.4). A further investigation into the results is then performed which highlights the benefits of the approach in general and looks at where future improvements can be made e.g. viewing misclassifications (Section 4.6). Finally, a summary and discussion of our evaluation is given (Section 4.7).

### 4.1 Evaluation methodology

In our evaluation, we employ two training strategies: 1) **One Category** training approach and 2) **Multiple Category** training approach i.e. MTL (as shown in Fig 7). We use these different approaches at various stages in the evaluation and finally compare them at the end of this section. When generating results for the One Category approach, we use the same category data to train and test the model e.g. train and test on dog food. In the Multiple Category approach, we train one model using a large number of product categories and then test individually on categories the model has seen during training and not seen during training (as described in Section 3.5) e.g. train on dog food, nail polish, shampoo, etc. and test on a Seen Training Category like dog food as well as an Unseen Category like cookies. Although we use the same product category data for the One Category and Multiple Category approaches, this data is still split to remove any train/test overlap.

As discussed in Section 3.3, as a result of the 36-month (i.e. length of Previous Time Window) rolling window nature of our approach, instances with the same keyphrase for the same product category between Fixed Time Windows (i.e. months) are highly similar e.g. the two instances of "charcoal" in the product category Toothpaste for the months 2014-05-01 and 2014-06-01 are nearly the same. As a result of this data overlap, we are constrained to having the training and testing data for the One Category approach separated by at least 36 Previous Time Windows (i.e. 36 months) because we use 36 Previous Time Windows of data for each instance (to avoid potential train/test contamination). We have multivariate time series data available from 2014-01-01 to 2018-12-31. Due to this train/test overlap issue, we train on data from 2014-01-01 to 2014-12-31 and test on data from 2018-01-01 to 2018-12-31. For similar reasons, we follow the same train/test split for the Multiple Category approach. As discussed in Section 3.1 and Section 3.5, this is the reason why 7 (and not 15) categories are used in the model training process. 7 categories have ground truth data at/before the dates between 2014-01-01 to 2014-12-31 (see Table 2). These are the only categories used in the One Category

approach in our experiments, as the One Category uses the same category to train and test the model i.e. if no training data is available for a category then it cannot be tested. These are also the only categories used to train the Multiple Category MTL approach, therefore making up the Seen Testing Categories (described in Section 3.5). The remaining 8 categories are solely used to test the Multiple Category MTL model to see if it generalizes on categories it has not seen during training, thus making up the Unseen Testing Categories (described in Section 3.5).

As discussed in Section 3.5, we use a Mini Rocket model followed by a Linear Ridge Regression classifier when detecting future customer needs. During the model-building process, we discovered that these two models are infeasible to use on the entirety of our training data from a time complexity standpoint. This is because on average each category has $\sim$ 34,000 instances every month with 1,263 univariate time series, which are each 36 months in length. This results in $\sim$ 18,550,944,000 data points for each category over 12 months i.e. from 2014-01-01 to 2014-12-31. We do not have the time or computing power to deal with this data, even with the fast speed of Mini Rocket. Hence, we undersample our data. This significantly reduces the number of instances used for training as there is a massive data imbalance between the positive and negative ground truth labels (Section 3.5). Specifically, there is a $\sim$ 260:1 ratio of negative instances to positive instances before undersampling across the categories used in the analysis from 2014-01-01 to 2014-12-31. When undersampling, we employ random undersampling—https://imbalanced-learn.org/stable/references/generated/imblearn.under_sampling. RandomUnderSampler.html.

For our task, undersampling allows us to significantly reduce training time, however, it comes with the drawback of not representing the true distribution of the output class(es) which affects predictions produced by classifiers [133]. For example, a model trained on all the training data versus a model trained on an undersampled version of the same data could predict a different output class for the same instance. Our model likely predicts far too many instances as the positive class (i.e. future customer needs) during testing given that it's trained on a downsampled version of the data which highly undersampled the negative class. To mitigate this undesired consequence, we use the information from the predicted probability output of the model rather than asking it what class an instance belongs to. We use this predicted probability output to find an optimal threshold that yields the highest performance on a held-out validation set at predicting future customer needs e.g. all instances that have a probability output greater than 0.8 (threshold) for the positive class are predicted as positive instances. Specifically, we validate the probability output of the Linear Ridge Regression classifier (which is first trained on the 2D embeddings produced by the Mini Rocket algorithm run over the 3D multivariate time series data). When choosing the threshold, we perform an exhaustive grid search for the probability value which yields the best F1 score between the values 0 to 1 with a step size of 0.01 (we elaborate on the choice of the F1 metric for model performance later in this section). It's noteworthy that finding this probability threshold for the classification of imbalanced data is an area that has been thoroughly explored in the ML literature [134]. We also note that we don't validate any other parameters in the overall process for computation/ time purposes. Such parameters may include the inputs into Mini Rocket (e.g. number of kernels) or Linear Ridge Regression (e.g. alpha parameter).

When splitting the training data from 2014-01-01 to 2014-12-31 into training and validation sets we don't use the traditional approach of randomly splitting the data on a per-instance basis, as instances with the same keyphrase could have highly similar multivariate time series (as previously discussed in this section). This would not represent the real relationship between the training and testing sets, which have no overlap. To bypass this issue, we split on the unique keyphrase level instead which ensures no training/validation data overlap. Specifically, we randomly sample 90% of the unique keyphrases from the training data and use the

instances associated with these keyphrases to train the model. The remaining 10% of keyphrases representing instances are used to validate it. When splitting the keyphrases, we use stratified random sampling—where unique keyphrases are divided based on if they ever become a future customer need i.e. contained in the TCN dataset 1–3 years in the future. When training the model, the data is then undersampled to a 1:1 ratio of positive and negative instances (allowing for faster training times). The validation data is then only undersampled to the ratio which represents the initial class distribution between positive and negative in the original training data e.g. 260:1 ratio. This is done as a probability threshold which represents the class distribution in the real training data is required to be estimated. By removing 90% of the unique keyphrases that represent future customer need instances this distribution is misrepresented, hence the need to undersample again.

**Social Media Algorithm**: How the One Category approach is trained, validated and tested

```
Data: catgoryArr ← ['shampoo', 'toothpaste', 'eyeliner']
for category ∈ categoryArr do
  allData = getDataForCategory(category)
  allTrainData, allTestData = splitDataByDate(allData)
  trainData, validData = splitDataByValidProcedure(allTrainData)
  trainDataUndersampled = undersample(trainData)
  model = trainModel(trainDataUndersampled)
  probabilityThreshold = validate(model, validData)
  prediction = test(allTestData, model, probabilityThreshold)
end
```

The Social Media Algorithm shows the pseudocode for how we validate, train and test our model. It draws upon all of the topics discussed in this section e.g. how we undersample our data, how we split our data into training/validation sets etc. The Social Media Algorithm specifically shows how we do this for the One Category approach (i.e. train and test using the same product category), however, this example can easily be extended to the Multiple Category approach by instead using multiple categories to train/validate/test the model. To start, we define the product categories we analyze i.e. *categoryArr*. In the algorithm, these categories are shampoo, toothpaste and eyeliner. However, in our experiments, we analyze more than just these categories, as discussed throughout the study. These categories are looped through with various processes being applied in each iteration which are responsible for training, validating and testing the model. For the first process applied in each iteration, we get all the data for the category being analyzed i.e. **getDataForCategory**. We assume all processing has been applied in this step to turn the Reddit posts into candidate keyphrases—which each consist of multiple features in the form of univariate time series (as described in Section 3.3). Secondly, we split the multivariate time series data by date into training and testing splits i.e. **splitDataByDate**. Due to the train/test overlap issue described in this section, for each category, we reserve data from 2014-01-01 to 2014-12-31 for training and 2018-01-01 to 2018-12-31 for testing. Thirdly, we further split the training data into training and validation sets i.e. **splitDataByValidProcedure**. Here we split on the unique keyphrase level (for the reasons described in this section). We then undersample the training data, using random undersampling, due to it being computationally infeasible to run the model on the entirety of our data (as described in this section) i.e. **undersample**. The model (Mini Rocket followed by a Linear Ridge Regression classifier) is then trained on the undersampled data i.e. **trainModel**. It's then validated on the held-out validation set to obtain the probability threshold which optimises the F1 score i.e. **validate**. This is done as the output class distribution represented by the undersampled training data does not estimate the class distribution of the testing data (which is not sampled). When the model is finally applied to the testing data (i.e. **test**), we use the probability threshold when classifying instances e.g. if the probability output of the instance is above the 0.8 threshold it's classified as a positive instance.

When running our experiments on the test data we noticed that different versions of the same model type were sometimes not producing similar results e.g. One Category model for lip balm not producing the same results. This is due to the various stochastic processes we perform when transforming our data. Such processes include applying the Mini Rocket algorithm which produces random 2D kernel embeddings for each run or undersampling our training data randomly. So to obtain a realistic result set, we run each approach 10 times and report the mean results in our experiments. This increases the time complexity of our experiment, however, reduces experimental bias.

## 4.2 Evaluation approaches

When assessing our approach we use two evaluation strategies. The first evaluates the model on the instance level in a binary classification setting, thus assessing the performance of the model on instances it predicts as future customer needs. The second evaluates the model based on the lists of ranked keyphrases it produces each Fixed Time Window (i.e. month), therefore assessing the model's capability of creating lists of keyphrases it predicts as future customer needs.

**4.2.1 Binary Classification Evaluation.** For the first approach, we simply evaluate on the instance level for the binary task of predicting whether the keyphrase instance will become a trend in 1–3 years in the TCN dataset. With our task being an imbalanced classification problem, the F1 measure is used. This is used as accuracy is not a suitable metric for these types of problems [135] e.g. could predict everything as the majority class and still achieve high accuracy. The F1 metric is a trade-off between precision and recall, both suitable metrics for the evaluation of our task (and imbalanced classification tasks in general [135]). Recall is necessary for the task as it evaluates the number of future customer needs that can be found by the model while precision is needed to keep the number of false positives low i.e. everything can't be predicted as a future customer need.

**4.2.2 List Evaluation.** The second approach we use evaluates a ranked list of submitted keyphrases for each Fixed Time Window (i.e. month). We use this procedure as it's the exact evaluation used by the baseline approach [35] (detailed in Section 4.3). The baseline approach itself informed its evaluation procedure based on related text mining literature [107, 136] for which many of its decisions for assessment are made. We also use this as we want to show that our approach performs better even when using the same evaluation methodology as the baseline. Although we use the same evaluation approach, we do not use the same evaluation data as the baseline. This is because the baseline only uses evaluation data for one category in its experiments i.e. Toothpaste. We instead use the aforementioned TCN dataset which incorporates multiple categories and is in the same final output as the evaluation data used by the baseline i.e. lists of keyphrases representing customer needs addressed in real products. To assess our approach using this ranked List Evaluation, we transform the output of our classification approach to a ranked list of keyphrases for each Fixed Time Window (discussed in Section 4.3). This is also done for the baseline approach, which produces a ranked list of keyphrases so that it can be assessed by the Binary Classification Evaluation (also discussed in Section 4.3).

Differently from the Binary Classification Evaluation, when comparing ranked lists of keyphrases in each Fixed Time Window, this approach considers string matches between the output submitted by the model and the TCN dataset within a Levenshtein distance of 0.8 [35]. This is done to allow for some potential misspellings that can occur on social media. As the keyphrases in this approach are submitted in ranked lists, this is done over reduced numbers of submitted keyphrases by the model run over the UGC data and the TCN dataset. Specifically, the number of keyphrases $K$ used is 5, 10, 15, and 20 (as in the baseline [35]). This is enough to capture highly important needs (e.g. top 5 needs) and also needs that are slightly

less important but still relevant (e.g. top 20 needs). In this evaluation, two metrics are recorded a) **List Mean Precision** and b) **List Recall**. To calculate List Mean Precision, List Precision is first calculated. List Precision is calculated at each Fixed Time Window (i.e. each month) and is defined as the number of correct keyphrases the model run over UGC can find (1–3 years ahead of the date in which the keyphrases are found) divided by the number of keyphrases $K$ it produced. As List Precision is calculated every month, List Mean Precision over the entire evaluation period can be computed, which is defined as the mean of the List Precision scores. List Recall is defined as the total number of unique keyphrases the model run over UGC can match in the TCN dataset (i.e. 1–3 years ahead) divided by the total unique number of keyphrases $K$ the TCN dataset produces in the entirety of the testing period. This metric instead focuses on assessing the unique customer needs detected by the model which are contained in the TCN dataset. However, the model run over UGC produces keyphrases from 2018-01-01 to 2018-12-31 and the ground truth data produces keyphrases from 2019-01-01 to 2021-12-31 in our experiments. As pointed out in the baseline study [35], this makes it very difficult to achieve high performance for this metric (i.e. List Recall).

## 4.3 Baseline approach comparison

In this section, we outline how our model performs against a baseline [35] using the two evaluation approaches defined in Section 4.2 i.e. Binary Classification Evaluation and List Evaluation. Here we only compare the baseline against the One Category approach i.e. training/testing on the same product category (Section 4.1). We do this as we want to clearly show that the ML approach is better than the baseline, to illustrate that future customer needs can be predicted with better performance than previous approaches i.e. our first research question discussed at the beginning of our evaluation. In Section 4.4, we show the performance of the Multiple Category MTL approach.

The baseline approach we compare against is a recent rule-based algorithm that finds customer need keyphrases from Reddit that are of interest to businesses [35]. It addresses the same overall task as in our study i.e. predicts candidate keyphrases which it estimates will be future customer needs addressed in real products. However, these keyphrases are predicted in a ranked list rather than a binary output (as in this study). It does this by performing 3 main steps, each containing many substeps: 1) data reduction; 2) keyphrase extraction; and 3) keyphrase ranking. For the final step (i.e. keyphrase ranking), it's noteworthy that the algorithm incorporates knowledge from Google Trends, as well as Reddit, as it leads to an increase in performance. The baseline approach is evaluated on the task of identifying future customer needs in Toothpaste products [35]. It's able to significantly outperform a random baseline for both the metrics discussed in Section 4.2 i.e. List Mean Precision and List Recall. It's also able to detect 4 out of 6 highly important needs identified by a large Multinational Corporation (MNC) specializing in the oral-care sector. Even though the baseline is only assessed for one product category in its evaluation, we compare it for three categories as our study is a multi-category analysis. These categories are 1) Toothpaste, 2) Perfume and 3) Dog Food. We don't compare for more than these categories due to the time it takes to collect the Google Trends data for each category for the baseline to work. When picking categories to compare, we chose Toothpaste as it's the one used in the baseline. We selected the other two categories as they make up a diverse range of categories for the baseline comparison experiment. There are several parameters required to run the baseline, including category-specific input parameters. We have intended to act as favourably as possible to the baseline when providing it with these parameters for categories it didn't use in its evaluation i.e. Dog Food and Perfume. S11 Appendix further details the parameters and parameter values used for each category.

**4.3.1 Baseline approach comparison—Binary Classification Evaluation.** As the output of the rule-based baseline is a ranking of keyphrases for each Fixed Time Window (i.e. month), some manipulations are required for it to be able to be assessed using the Binary Classification Evaluation (as discussed in Section 4.2). To turn this ranking approach into a form suitable for binary classification, we use the natural ordering of the keyphrases when classifying. Specifically, we use a threshold to set the first number of ranked keyphrases (sorted in ascending order of rank) to be true (i.e. are future customer needs) while the others remain false (i.e. are not future customer needs). In our experiments, we use 15 of these thresholds to explore which one yields the best F1 score. Precisely, we use the thresholds 5, 10, 15, 20, 50, 100, 250, 500, 750, 1000, 1250, 1500, 1750, 2000, and 2500. These thresholds are used as they represent a large and widespread range of values used to find a near-optimal F1 for the baseline. Smaller increment values are used at the start (e.g. 5, 10, 15, 20) due to there being only a small number of positive instances in the dataset (Section 4.1).

The results of this evaluation for the One Category approach and baseline approach are seen in Table 5. As seen in the table, the One Category approach outperforms the best baseline approach across all 3 categories. Furthermore, it outperforms the baseline on the categories that aren't addressed in its experiments by a large margin i.e. Dog Food and Perfume. This is probably because the baseline's parameter values aren't tailored to these categories. In the broader picture, however, it does better because it uses supervised ML. It therefore can learn from past instances of future customer needs which generalizes better than rules encoded by a human in the baseline approach e.g. is the *Min Chi Square P-value* parameter $< 0.02$? For completeness, we also record the precision and recall results of this comparison (S12 Appendix).

**Table 5. Binary Classification Evaluation showing the mean F1 scores (rounded to 3 decimal places) for the One Category and baseline approaches.** The best result for the baseline across each threshold for each category is in bold.

| Category | Method | | | | |
|---|---|---|---|---|---|
| | *One Category* | *Baseline* | | | |
| Toothpaste | 0.099 | *5* | *10* | *15* | *20* | *50* |
| | | 0.012 | 0.016 | 0.023 | 0.025 | 0.036 |
| | | *100* | *250* | ***500*** | *750* | *1000* |
| | | 0.049 | 0.069 | **0.073** | 0.072 | 0.071 |
| | | *1250* | *1500* | *1750* | *2000* | *2500* |
| | | 0.068 | 0.064 | 0.059 | 0.057 | 0.055 |
| Dog Food | 0.120 | *5* | *10* | *15* | *20* | *50* |
| | | 0.000 | 0.000 | 0.003 | 0.006 | 0.019 |
| | | *100* | *250* | *500* | *750* | *1000* |
| | | 0.028 | 0.057 | 0.065 | 0.068 | 0.069 |
| | | ***1250*** | *1500* | *1750* | *2000* | *2500* |
| | | **0.070** | 0.068 | 0.066 | 0.061 | 0.059 |
| Perfume | 0.130 | *5* | *10* | *15* | *20* | *50* |
| | | 0.000 | 0.001 | 0.002 | 0.006 | 0.035 |
| | | *100* | *250* | *500* | *750* | ***1000*** |
| | | 0.052 | 0.063 | 0.071 | 0.080 | **0.085** |
| | | *1250* | *1500* | *1750* | *2000* | *2500* |
| | | 0.085 | 0.080 | 0.076 | 0.070 | 0.059 |

The mean F1 scores (rounded to 4 decimal places) for Perfume at the thresholds 1000 and 1250 are 0.0852 and 0.0850 respectively, hence why the threshold at 1000 is the best baseline.

In this section, we also compare the F1 scores of the One Category and baseline approaches using a statistical test for each of the product categories used in the baseline comparison i.e. Toothpaste, Dog Food and Perfume. This can be run as both the One Category approach and the baseline approach are run multiple times, as detailed in Section 4.1 and Section 4.3. Specifically, a Mann-Whitney U test [137] is run comparing the F1 scores of the One Category approach and the "best" baseline approach. The "best" baseline approach for each category is represented by the threshold which has the highest mean performance seen in Table 5 i.e. 500 for Toothpaste, 1250 for Dog Food and 1000 for Perfume. As with previous studies comparing result data from the output of different ML models [35, 72], the reason this test is used instead of a t-test is that the F1 scores for each approach are not normally distributed [137]. The t-test compares the means of the two samples and assumes they are normally distributed while the Mann-Whitney U test compares the rank sum of the two samples and doesn't assume they are normally distributed [137]. It's important to note, that for the same reasons, this test is used throughout our evaluation to compare different samples of results. Table 6 shows the p-value (rounded to 3 decimal places) of this test for each product category. If the test is in favour of the baseline by the median scores being greater than the One Category scores, a + is post-fixed to the result (as in [72]). The reason why we compare the median scores, in this case, is because the Mann-Whitney U test (i.e. test we use to compare results) is a "test of medians" [138]—therefore the test is in favour of the baseline if the median scores are greater than the One Category scores. The results in the table solidify the fact that the One Category approach is better than the baseline for the F1 metric using the Binary Classification Evaluation. This is because for all the categories analyzed the results from the two samples are significantly different (i.e. all p-values are $<0.001$) and the median results for the One Category approach are all greater than the baseline.

**4.3.2 Baseline approach comparison—List Evaluation.** Similarly to the output of the ranking algorithm, the proposed ML approach addressed in this study needs to have its output transformed for it to be evaluated by the List Evaluation approach i.e. to calculate List Mean Precision and List Recall (described in Section 4.2). Specifically, this involves changing the binary prediction output to a ranked list of keyphrases each Fixed Time Window (i.e. month). This is done by using the predicted probability score outputted by the Linear Ridge Regression classifier (i.e. ML model used in this study) to rank the terms of each Fixed Time Window in descending order of confidence. By ranking this way, the instances the model estimates are most likely to become future customer needs will be at the top of the list, while the ones it least estimates will become future customer needs will be at the bottom.

The results of the evaluation for the One Category and baseline approach are seen in Table 7. The One Category approach is better than the baseline across all the results for the Dog Food and Perfume categories. However, the baseline performs better for the Toothpaste category by obtaining higher performance on all the List Mean Precision results and one of the List Recall results. As discussed previously in the evaluation, the baseline is specifically tuned

**Table 6. P-value (rounded to 3 decimal places) for the Mann-Whitney U test of F1 scores from the One Category approach vs the best baseline approach for Binary Classification Evaluation.**

| Category | F1 |
|---|---|
| Toothpaste | $<0.001$ |
| Dog Food | $<0.001$ |
| Perfume | $<0.001$ |

+ test is in favour of the Baseline

**Table 7. List Evaluation showing the mean results (rounded to 3 decimal places) for the One Category and baseline approaches.** For each category, the result from the best approach is in bold.

| Category | Method | Metric | | | | | | | |
|---|---|---|---|---|---|---|---|---|---|
| | | Mean Precision | | | | Recall | | | |
| | | K = 5 | K = 10 | K = 15 | K = 20 | K = 5 | K = 10 | K = 15 | K = 20 |
| Toothpaste | One Category | 0.057 | 0.109 | 0.124 | 0.155 | 0.009 | **0.026** | **0.038** | **0.054** |
| | Baseline | **0.221** | **0.205** | **0.19** | **0.161** | **0.011** | 0.015 | 0.019 | 0.023 |
| Dog Food | One Category | **0.11** | **0.159** | **0.203** | **0.225** | **0.011** | **0.029** | **0.042** | **0.057** |
| | Baseline | 0.0 | 0.0 | 0.033 | 0.049 | 0.0 | 0.0 | 0.007 | 0.01 |
| Perfume | One Category | **0.102** | **0.179** | **0.205** | **0.244** | **0.026** | **0.05** | **0.076** | **0.104** |
| | Baseline | 0.0 | 0.005 | 0.009 | 0.018 | 0.0 | 0.006 | 0.012 | 0.023 |

for the Toothpaste category across the metrics used in the List Evaluation, so it's not surprising that it performs better here.

As in Section 4.3.1, we also compare the results of the One Category and baseline approaches using a Mann-Whitney U test, as they are run multiple times. Table 8 shows the p-value (rounded to 3 decimal places) for each metric in the List Evaluation over every product category used in the baseline comparison. For all the results in the table, the One Category approach is significantly better 18/24 times when the p-value level is either 0.1, 0.05 or 0.01 (i.e. test is in favor of the One Category approach and the p-value is under the mentioned levels). The baseline is significantly better 4/24 times when the p-value level is 0.1 and 3/24 times when the level is 0.05 or 0.01. The levels (i.e. 0.1, 0.05, 0.01) are reported here as they have been commonly used in other studies to test for statistical significance [139]. The results in this table demonstrate that the One Category approach is better than the baseline for the List Evaluation barring the List Mean Precision metric for the Toothpaste category.

**4.3.3 Baseline approach comparison—Summary.** To summarize, the One Category approach outperforms the baseline entirely in the Binary Classification Evaluation (Section 4.3.1) and mostly in the List Evaluation (Section 4.3.2), except for the List Mean Precision metric for the Toothpaste category. Considering these results, we address our first research question that future customer needs can be predicted with better performance than previous approaches using UGC (as discussed at the beginning of our evaluation).

## 4.4 Impact of Multi-Task Learning

In this section, we outline how the Multiple Category approach achieves similar performance to the One Category approach i.e. MTL model trained on all product categories is similar to

**Table 8. P-value (rounded to 3 decimal places) for the Mann-Whitney U Test of results from the One Category approach vs the baseline approach for List Evaluation.**

| Category | Metric | | | | | | | |
|---|---|---|---|---|---|---|---|---|
| | Mean Precision | | | | Recall | | | |
| | K = 5 | K = 10 | K = 15 | K = 20 | K = 5 | K = 10 | K = 15 | K = 20 |
| Toothpaste | <0.001+ | <0.001+ | 0.001+ | 0.877+ | 0.086+ | 0.249 | 0.002 | <0.001 |
| Dog Food | <0.001 | <0.001 | <0.001 | <0.001 | <0.001 | <0.001 | <0.001 | <0.001 |
| Perfume | <0.001 | <0.001 | <0.001 | <0.001 | <0.001 | <0.001 | <0.001 | <0.001 |

[+] test is in favour of the baseline

using the same category data to solely train/test the model. We do this as in the previous section (i.e. Section 4.3), we showed that our ML approach is better than the current baseline in the literature i.e. our first research question. Therefore, in this section we address the second research question (discussed at the beginning of our evaluation) that MTL can achieve similar detection performance for the classification of future customer needs to the One Category approach. This is important as an MTL model can be used to predict categories not seen in the training process, therefore generalizing to unseen categories without having to be retrained.

As discussed in Section 3.5, there are two types of categories used to test the MTL model: 1) Seen Testing Categories; and 2) Unseen Testing Categories. The Seen Testing Categories are categories used in the training process. Conversely, the Unseen Testing Categories are categories used in testing but are not used by the model in the training process. These Seen Testing Categories are also the only categories used in the One Category approach as they have data to train and test on the same category. As seen in Table 2, the Seen Testing Categories are Dog Food, Eyeliner, Lip Balm, Nail Polish, Perfume, Shampoo and Toothpaste. The Unseen Testing Categories make up the 8 remaining categories in Table 2 i.e. Beer, Cereal, Coffee, Cookie, Pizza, Popcorn, Soda and Soup. In this section, we carry out two separate evaluations for assessing the Seen and the Unseen Testing Categories. The main reason for this is that a comparison analysis between the One Category approach and the Multiple Category approach can only be performed on the Seen Testing Categories because the One Category approach can only be performed on these categories. Although the Unseen Testing Categories are not used in the comparison analysis (i.e. to examine whether the MTL approach is better than using the same category to train/test a model), they still contribute to the evaluation as they test if the MTL model is capable of detecting future customer needs on categories it hasn't seen during training e.g. can a model trained on Eyeliner, Toothpaste and Perfume predict an unseen category such as Cookies.

**4.4.1 Multi-Task Learning approach comparison for Seen Categories—Binary Evaluation.** In this section, we compare the One Category approach to the Multiple Category MTL approach for the Seen Testing Categories using the Binary Classification Evaluation (Section 4.2). The results of this evaluation, shown in Table 9, illustrate that the Multiple Category approach outperforms the One Category approach across 5 of the 7 categories. The One Category approach obtains higher performance in 1 category (i.e. Perfume) and they both achieve the same performance for 1 category (i.e. Shampoo). The precision and recall scores associated with the F1 scores in Table 9 are also recorded (S12 Appendix).

As in Section 4.3, we also compare the F1 scores of the One Category and Multiple Category approaches for the Seen Testing Categories using a Mann-Whitney U test. Table 10 shows the

**Table 9. Binary Classification Evaluation showing the mean F1 scores (rounded to 3 decimal places) for the One Category and Multiple Category approaches across the Seen Testing Categories.** For each category, the result from the best approach is in bold.

| Category | Method | |
|---|---|---|
| | *One Category* | *Multiple Category* |
| Dog Food | 0.120 | **0.121** |
| Eyeliner | 0.069 | **0.077** |
| Lip Balm | 0.099 | **0.136** |
| Nail Polish | 0.096 | **0.116** |
| Perfume | **0.130** | 0.085 |
| Shampoo | **0.115** | **0.115** |
| Toothpaste | 0.099 | **0.115** |

**Table 10. P-value for the Mann-Whitney U Rank Test of F1 scores (rounded to 3 decimal places) from One Category approach vs Multiple Category approach for Binary Classification Evaluation across the Seen Testing Categories.**

| Category | F1 |
|---|---|
| Dog Food | 0.971 |
| Eyeliner | 0.247 |
| Lip Balm | 0.015 |
| Nail Polish | 0.19 |
| Perfume | 0.019+ |
| Shampoo | 0.853+ |
| Toothpaste | 0.247 |

[+] test was in favour of the One Category approach

p-value (rounded to 3 decimal places) of this test for each product category analyzed. Although the Multiple Category approach performs better across 5 of the 7 categories (as shown in Table 9), it only performs significantly better 1/7 times when the p-value level is 0.1 or 0.05 and never when the level is 0.01. It's of note that the Multiple Category approach for Shampoo is the same as the One Category approach in Table 9, however, the One Category approach slightly outperforms it in Table 10. This is because the mean result is recorded in Table 9 and the median is recorded in Table 10. The baseline is also better 1/7 times when the p-value level is 0.1 or 0.05 and never when the level is 0.01. The results in the table show that the Multiple Category approach performs similarly to the One Category approach for the Binary Classification Evaluation.

**4.4.2 Multi-Task Learning approach comparison for Seen Categories—List Evaluation.** In this section, we compare the One Category approach to the Multiple Category MTL approach for the Seen Testing Categories using the List Evaluation (discussed in Section 4.2). As performed in the baseline comparison (i.e. Section 4.3), we change the output of both the One Category and Multiple Category approaches for them to be evaluated using the List Evaluation approach. The results of this evaluation are seen in Table 11. The One Category outperforms the Multiple Category approach, with it obtaining 33 of the best results from a total of 56 in the table. The Multiple Category approach obtains 21 of the best results while they both obtain the same result twice (i.e. Recall when $K$ is 10 for Nail Polish and Recall when $K$ is 5 for Toothpaste).

As in Section 4.3, we also compare the results of the One Category and Multiple Category approaches using a Mann-Whitney U test. Table 12 shows the p-value (rounded to 3 decimal places) of this test for each product category. Although the results in Table 11 may portray that many of the results are better for the One Category approach, in fact, it only performs significantly better 6/56 times and 4/56 times when the p-value level is 0.1 and 0.05 and never when the level is 0.01. The Multiple Category approach also only performs significantly better 3/56 times and 2/56 times when the p-value level is 0.1 and 0.05 and never when the level is 0.01. Furthermore, for both approaches, statistical significance across the mentioned values is only ever achieved in 2 categories: Perfume (One Category) and Lip Balm (Multiple Category). The results in the table show that the Multiple Category approach performs similarly to the One Category approach for the List Evaluation. To summarize, the Multiple Category approach performs similarly to the One Category approach in the Binary and List Evaluation approaches. Due to this, we partly address our second research question that future customer needs can be predicted with similar performance using MTL (as discussed at the beginning of our evaluation).

**Table 11. List Evaluation showing the mean results (rounded to 3 decimal places) for the One Category and Multiple Category approaches across the Seen Testing Categories.** For each category the best approach is in bold.

| Category | Method | Metric | | | | | | | |
|---|---|---|---|---|---|---|---|---|---|
| | | Mean Precision | | | | Recall | | | |
| | | K = 5 | K = 10 | K = 15 | K = 20 | K = 5 | K = 10 | K = 15 | K = 20 |
| Dog Food | One Category | **0.110** | **0.159** | **0.203** | **0.225** | **0.011** | **0.029** | **0.042** | **0.057** |
| | Multiple Category | 0.055 | 0.122 | 0.161 | 0.178 | 0.008 | 0.028 | 0.039 | 0.052 |
| Eyeliner | One Category | **0.058** | **0.085** | **0.108** | 0.111 | **0.015** | **0.028** | **0.041** | 0.053 |
| | Multiple Category | 0.037 | 0.059 | 0.099 | **0.117** | 0.007 | 0.019 | 0.037 | **0.054** |
| Lip Balm | One Category | 0.092 | 0.153 | 0.174 | 0.194 | 0.014 | 0.031 | 0.040 | 0.052 |
| | Multiple Category | **0.140** | **0.167** | **0.217** | **0.230** | **0.025** | **0.038** | **0.065** | **0.078** |
| Nail Polish | One Category | 0.120 | 0.165 | 0.186 | 0.183 | 0.021 | **0.044** | 0.060 | 0.074 |
| | Multiple Category | **0.143** | **0.185** | **0.197** | **0.190** | **0.024** | **0.044** | **0.063** | **0.083** |
| Perfume | One Category | **0.102** | **0.179** | **0.205** | **0.244** | **0.026** | **0.050** | **0.076** | **0.104** |
| | Multiple Category | 0.042 | 0.076 | 0.107 | 0.126 | 0.016 | 0.031 | 0.045 | 0.063 |
| Shampoo | One Category | **0.043** | **0.096** | **0.170** | **0.187** | **0.014** | **0.028** | **0.054** | **0.066** |
| | Multiple Category | 0.033 | 0.058 | 0.142 | 0.148 | 0.009 | 0.018 | 0.047 | 0.059 |
| Toothpaste | One Category | 0.057 | 0.109 | 0.124 | 0.155 | **0.009** | **0.026** | **0.038** | **0.054** |
| | Multiple Category | **0.073** | **0.132** | **0.156** | **0.183** | **0.009** | 0.021 | 0.033 | 0.053 |

**4.4.3 Multi-Task Learning approach comparison for Unseen Categories—Binary Evaluation.** In this section, we show the results of the Multiple Category MTL approach for the Unseen Categories using the Binary Evaluation (discussed in Section 4.2). The results of this evaluation are seen in Table 13. We also record the precision and recall scores associated with the F1 results in Table 13 (S12 Appendix). It would be an unfair test to compare the results from the Seen and Unseen Testing Categories using a statistical test because some categories are predicted with better performance than others—therefore making the test unfair. That being said, the results in the table are not too different from the Seen Testing Category results in Table 9. This shows that (according to the Binary Evaluation) the MTL model can still predict future customer needs on a category it has not seen during training with relatively similar performance to ones it has seen during training. This is very useful because even if there is no ground truth category data available for a product category, future customer needs for it can

**Table 12. P-value (rounded to 3 decimal places) for the Mann-Whitney U Test of results from One Category approach vs Multiple Category approach for List Evaluation.**

| Category | Metric | | | | | | | |
|---|---|---|---|---|---|---|---|---|
| | Mean Precision | | | | Recall | | | |
| | K = 5 | K = 10 | K = 15 | K = 20 | K = 5 | K = 10 | K = 15 | K = 20 |
| Dog Food | 0.123+ | 0.283+ | 0.393+ | 0.878+ | 0.393+ | 0.82 | 0.436+ | 0.939+ |
| Eyeliner | 0.739+ | 0.49+ | 0.315+ | 0.444+ | 1.0 | 0.939+ | 0.796 | 0.82+ |
| Lip Balm | 0.123 | 0.082 | 0.912+ | 0.401+ | 0.393 | 0.012 | 0.353 | 0.014 |
| Nail Polish | 0.684 | 0.785+ | 0.739 | 0.789+ | 0.796 | 0.789+ | 0.971 | 0.85+ |
| Perfume | 0.123+ | 0.229+ | 0.043+ | 0.046+ | 0.052+ | 0.047+ | 0.029+ | 0.073+ |
| Shampoo | 0.796+ | 0.599+ | 0.315+ | 0.222+ | 0.436+ | 0.47+ | 0.436+ | 1.0 |
| Toothpaste | 0.529 | 0.656 | 0.684 | 0.909+ | 0.529 | 0.703+ | 0.631+ | 0.88+ |

[+] test was in favour of the One Category approach

**Table 13. Binary Classification Evaluation showing the mean F1 scores (rounded to 3 decimal places) for the Multiple Category approach for the Unseen Testing Categories.**

| Category | Method |
|---|---|
| | *Multiple Category* |
| Beer | 0.086 |
| Cereal | 0.107 |
| Coffee | 0.086 |
| Cookie | 0.084 |
| Pizza | 0.118 |
| Popcorn | 0.109 |
| Soda | 0.070 |
| Soup | 0.083 |

still be predicted on Reddit. To further emphasise the fact that the results from the Seen and Unseen Testing Categories don't differ much from each other, we plot the distribution of F1 scores for these category types using the Multiple Category approach in S13 Appendix.

**4.4.4 Multi-Task Learning approach comparison for Unseen Categories—List Evaluation.** In this section, we show the results of the Multiple Category MTL approach for the Unseen Categories using the List Evaluation (discussed in Section 4.2). The results of this evaluation are seen in Table 14. As with the previous section (i.e. Section 4.4.3), it would not be fair to compare the results from the Seen and Unseen Testing categories using a statistical test, however, the results are not too dissimilar from the Seen Testing Category results in Table 11. Because the information seen in Tables 11 and 14 can be difficult to summarize, we also show the mean result across all the categories of the Multiple Category approach for each of the 80 Unseen Testing Categories and the 70 Seen Testing Categories results for each metric across each value of *K* in Table 15. As seen in the tables, the performance for the Seen and Unseen Testing Categories are very similar. This shows that (according to the List Evaluation) the MTL model can still predict future customer needs on a category it has not seen during training with relatively similar performance to ones it has seen during training. To further visualize the fact that the results from the Seen and Unseen Testing Categories don't differ much from each other, we plot the distribution of Mean Precision and Recall scores across all the mentioned values of *K* for these categories types using the Multiple Category approach in S14 Appendix. This in tandem with the Binary Evaluation for Unseen Testing Categories (i.e.

**Table 14. List Evaluation showing the mean results (rounded to 3 decimal places) for the Multiple Category approach across the Unseen Testing Categories.**

| Category | Method | Metric | | | | | | | |
|---|---|---|---|---|---|---|---|---|---|
| | | *Mean Precision* | | | | *Recall* | | | |
| | | *K = 5* | *K = 10* | *K = 15* | *K = 20* | *K = 5* | *K = 10* | *K = 15* | *K = 20* |
| Beer | Multiple Category | 0.075 | 0.097 | 0.120 | 0.125 | 0.016 | 0.032 | 0.052 | 0.068 |
| Cereal | Multiple Category | 0.052 | 0.082 | 0.125 | 0.131 | 0.007 | 0.020 | 0.036 | 0.043 |
| Coffee | Multiple Category | 0.107 | 0.120 | 0.136 | 0.141 | 0.017 | 0.038 | 0.052 | 0.064 |
| Cookie | Multiple Category | 0.040 | 0.119 | 0.149 | 0.163 | 0.014 | 0.040 | 0.071 | 0.091 |
| Pizza | Multiple Category | 0.075 | 0.110 | 0.143 | 0.159 | 0.013 | 0.027 | 0.045 | 0.057 |
| Popcorn | Multiple Category | 0.088 | 0.133 | 0.126 | 0.142 | 0.014 | 0.033 | 0.047 | 0.066 |
| Soda | Multiple Category | 0.090 | 0.118 | 0.128 | 0.126 | 0.022 | 0.040 | 0.062 | 0.075 |
| Soup | Multiple Category | 0.072 | 0.155 | 0.172 | 0.180 | 0.012 | 0.031 | 0.043 | 0.060 |

**Table 15. Seen and Unseen Testing Category mean results across all the categories used in the analysis for List Evaluation (rounded to 3 decimal places).** For each metric the result from the best approach is in bold.

| Category | Method | Metric | | | | | | | |
|---|---|---|---|---|---|---|---|---|---|
| | | Mean Precision | | | | Recall | | | |
| | | K = 5 | K = 10 | K = 15 | K = 20 | K = 5 | K = 10 | K = 15 | K = 20 |
| Seen | Multiple Category | **0.075** | **0.014** | 0.114 | 0.029 | **0.154** | 0.047 | **0.167** | 0.063 |
| Unseen | Multiple Category | **0.075** | **0.014** | **0.117** | **0.033** | 0.137 | **0.051** | 0.146 | **0.066** |

Section 4.4.3) shows that even if no ground truth data is available for a category, future customer needs for it can still be predicted on Reddit.

**4.4.5 Multi-Task Learning approach comparison—Summary.** To summarize, the Multiple Category approach and One Category approach perform similarly when assessed on Seen and Unseen Testing categories. This addresses our second research question (discussed at the beginning of our evaluation), that MTL achieves similar performance to the approach that uses the same category data to train and test a model for predicting future customer needs. Although the two approaches perform similarly, we recommend using the Multiple Category MTL approach. This is because it can provide predictions for categories it hasn't seen during training.

## 4.5 Comparison to State of the Art methods

In this section, we carry out a State of the Art (SOTA) method comparison analysis using 4 case study examples from work related to this study. This is performed to show how our approach fits into the general customer needs mining literature by comparing it to SOTA methods.

First, we compare our approach to a method that mines current customer needs in the form of ranked keyphrases for specific product models [75]. This method differs from the approach described in this study in two main ways: 1) it mines product models (e.g. Coca-Cola, Haribo) while our approach mines product categories (e.g. soft drinks, sweets); and 2) it predicts current customer needs while our approach predicts future customer needs. The approach in [75] uses LDA to rank keyphrases according to the topics they are in. In a case study, it achieved precision results ranging from 12% to 38% for detecting customer needs from 4 models of automobiles (i.e. Toyota Prius, Tesla Model S, Honda Civic and Jeep Wrangler). These precision metrics were obtained by having humans manually read through the predictions to check if they were correct. Although it is an unfair comparison, our method achieves precision results higher than this method i.e. 19.7% to 51.3% as shown in S14 and S15 Tables in S12 Appendix. Such a comparison is unfair as each approach addresses a different task and has a different evaluation methodology. That being said, achieving similar metrics given the task of predicting current customer needs in comparison to future customer needs (more difficult task) shows the value of the work in this study.

The second example case study also mines current customer needs in the form of ranked keyphrases for specific product models [14]. It has a highly similar methodology to the previous example case study and also uses a LDA based model when ranking customer needs. In a case study on ranking customer needs from 4 mobile phone products (i.e. iPhone4, Samsung Galaxy S II, Motorola Droid RAZR and Sony Ericsson Xperia Play), it achieved precision results ranging from 0.1–0.62. As in [75], these precision metrics were obtained by having humans scan through the ranked lists. The approach in this study achieves similar precision scores, thus showing its usefulness as an approach.

The third case study is the same study that was used as the baseline comparison in Section 4.3. As stated, this approach mines future customer needs using a rule-based algorithm that is

run over Reddit data. The task it addresses is the exact same as the one addressed in this study i.e. predicting future customer needs for product categories as ranked lists of keyphrases. Due to this, it can be easily compared with our approach. As shown in Section 4.3, our approach significantly outperforms the rule-based algorithm in various evaluation metrics across 3 different product categories. The approach in this paper can therefore be seen as contributing to the literature for the work in this area (as it makes significant improvements).

Finally, our approach is compared to a case study method which predicts future customer needs as a regression problem [103]. Specifically, this approach uses a fuzzy time series method to predict the importance of customer needs addressed in an electric hairdryer product. Comparing this approach to the one addressed in this study is difficult to do, given they both use different methods i.e. one outputs lists of keyphrases (keyphrase ranking) while another predicts a continuous value for a keyphrase (regression). Due to these methodological differences, these approaches are instead compared based on their usefulness of being able to predict far into the future. The approach in [103] has been shown to provide high performance at predicting Google Trends data far into the future and it has also shown how it's useful for the purposes of product development. Similarly, our approach is shown to perform highly compared to other methods at predicting ranked lists of customer needs that are of importance far into the future i.e. 1–3 years.

## 4.6 Further examination of results

In this section, we perform a deeper analysis of the results and see where certain optimizations can be made to improve the model's capability. We specifically examine the MTL approach as it's the model we recommend using (detailed in Section 4.4.5). Therefore, the discussion in this section assumes that this model is applied. As the analysis performed in this section does not present the major outcomes of our approach (e.g. our model performs better than a baseline), we only present high-level findings in this section. Linked appendices back up the specific claims made in this section.

The first analysis looks at the **lead times the MTL model detects future customer needs** in trending products in the marketplace i.e. in the TCN dataset. The finding from this is that although the model detects a large number of customer needs with lead times of less than 5 months in advance, it also detects a lot of needs with lead times of up to 2 years. It does this by checking how far out the keyphrases are from the month of prediction to the date they first trend in the TCN dataset. Fig 8 shows a kernel density estimate plot of these lead times across all 15 categories in the analysis. Such lead times would be highly beneficial for companies to identify before these needs start to become popular in the marketplace. Refer to S15 Appendix for a more detailed description of this analysis e.g. details on how the plot is generated.

The second analysis explores the room for **future optimizations** of the approach. The main finding is that large performance increases can be made in the way in which the parameters are validated, in particular the probability threshold parameter. An area where this could be improved is how we split the data into training and validation sets, which is not conventional given the nuances of the training data having overlapping time series (detailed in Section 4.1). Based solely on the estimation of the probability threshold parameter, the model can predict categories with an increased F1 score of 2.1%-5.4%. Refer to S16 Appendix for a more detailed description of this analysis.

The third analysis shows **alternative visual plots of the performance of the model**, specifically the Receiver Operating Characteristic (ROC) Curve and the Precision-Recall (PR) Curve. This is done to provide a different view of our results other than the F1 score which is used extensively in our study. When displaying the plots we also show the performance of a random

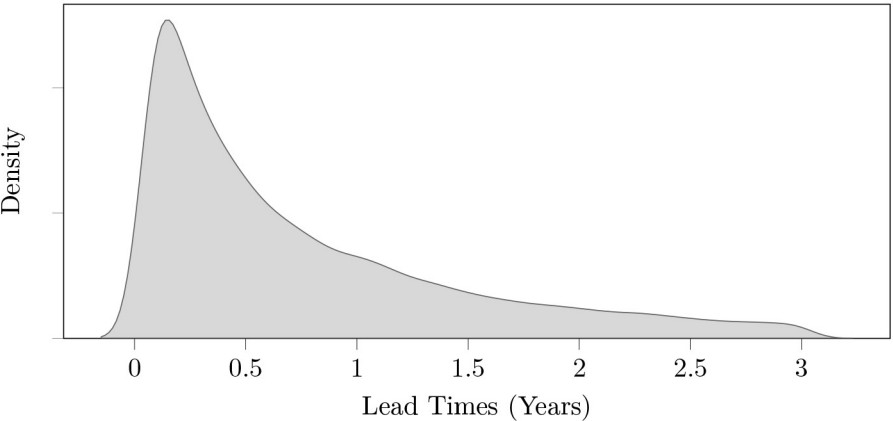

**Fig 8. Lead times (years) of detecting future keyphrase customer needs before they are addressed in the marketplace i.e. TCN dataset.**

classifier. The key finding reaffirms that our model achieves high performance given the task of predicting future customer needs. Refer to S17 Appendix for a more detailed description of this analysis.

The final analysis highlights the **misclassifications** made by the model. The main findings show the primary issue the MTL model makes is misclassifying keyphrases that are irrelevant for a particular product category e.g. "garlic" for the Beer category or "beef" for the Cookie category. This occurs due to how the MTL model is trained on a wide variety of product categories. Through training this way it learns the characteristics of a future customer need across multiple categories rather than needs for its own category. This is how the MTL approach obtains generalizable performance, however, it's also a drawback of the approach. Refer to S18 Appendix for a more detailed description of this analysis.

## 4.7 Summary and discussion

To help guide our evaluation we proposed two research questions at the beginning of the section: 1) can future customer needs be predicted with better performance than previous approaches using UGC from the social media platform Reddit across multiple product categories within CPG; and 2) can the use of MTL (described in Section 3.5) be employed to achieve similar performance to training/testing on the same category. To address these questions, we described how our approach is implemented and detailed the two training strategies used in our evaluation (Section 4.1): 1) One Category training—which uses the same category data to train and test the model at finding future customer needs; and 2) Multiple Category training—which incorporates MTL by using multiple categories to train a model. We also described the two evaluation approaches we used to test the described models (Section 4.2): 1) Binary Classification Evaluation and 2) List Evaluation.

Using the One Category approach, we compared our approach to a baseline in the literature (Section 4.3). We only used the One Category approach here as we wanted to test if our general ML approach is better than the baseline. We showed that our One Category model significantly outperformed the baseline in two of the described evaluation approaches across multiple categories used in the baseline analysis. By illustrating this we show that our approach can predict with better performance than previous approaches (question 1). The Multiple Category MTL approach was then tested against the One Category approach to observe if the approach

of training on multiple categories is similar to using one category to test the model. We showed that this was the case as the Multiple Category approach performed highly similar to the One Category approach in two of the evaluation approaches across multiple categories used in the experiment (question 2). By showing this we illustrate that the MTL model can be employed to predict on categories it hasn't seen, therefore being useful in situations where the category doesn't have ground truth training data available for it. It's noteworthy that the two research questions addressed in this evaluation map to two of the research contributions initially stated in our Introduction (Section 1): 1) task of predicting future customer needs is performed better than previous studies (question 1); and 2) the use of MTL is employed by incorporating data from multiple product categories to make predictions, which yields a model capable of predicting a category it doesn't use during training (question 2).

Throughout our evaluation some of the reported results may seem underwhelming drawing criticism e.g. F1 scores ranging from 7–14% for the best-performing MTL model across 15 product categories used to evaluate it. Similar criticisms are given to tasks with a high data imbalance which are also difficult to predict. Tasks of this kind are seen in a various range of topics in the ML literature including hashtag prediction [140], intrusion detection [141], image classification [142], predicting responses to intensive Post-Traumatic Stress Disorder (PTSD) treatment [143], predicting treatment discontinuation in patients with diabetes [144], classification of unstructured medical notes [145], etc. Depending on the level of imbalance these tasks can achieve similar performance to our study. In a particular study addressing the task of virality prediction of hashtags [140] with a class imbalance of 15:1, the best two models achieved an F1 score of 36.28% and an Area Under Curve (AUC) PR score of 30%. In a study on intrusion detection [141], a model achieved an AUC PR score of 20.51% at detecting blacklist intrusions with a label distribution ratio of ≈166:1. A classification model identifying images on Wikipedia achieved a mean F1 score of 26.7% across 31 labels which had a positive label percentage of 5.71–7.55% (depending on the dataset used). A model predicting Treatment Discontinuation (TD) for diabetes patients achieved an AUC PR score of 8.1%, 22.8% and 29% a respective 2, 3 and 4 months into treatment with a positive to negative label ratio of ≈30:1 in the training set and ≈26:1 in the test set. Finally, a text classification model predicting medical notes into 16 different classes obtained a mean AUC PR score between ≈10% and ≈90% depending on the prevalence of the disease (i.e. label distribution), with lower disease prevalence obtaining lower AUC PR scores. These studies show that models that perform difficult tasks with a high data imbalance generally achieve low performance (if evaluated correctly with high-quality ground truth data and suitable metrics). Although these studies build low-performing models they still perform useful tasks e.g. predicting responses to intensive PTSD treatment [143]. The same can be said for the area of research addressed in this study (i.e. predicting future customer needs) which has been talked about in various studies in the business literature for many years [46, 146–148].

## 5 Conclusion & future work

There are many families of approaches that mine customer needs from UGC. Some perform document classification by reducing the number of documents under analysis to ones that contain customer needs. Some cluster keyphrases or documents into groups that contain customer needs. Finally, there is a large body of work that focuses on the keyphrase level, by highlighting important ones that are deemed to be customer needs. From the literature analyzing on the keyphrase level, there is a lack of research addressing unmet customer needs through the form of predicting future ones. There is also an absence of supervised approaches that highlight important customer needs due to the unavailability of a ground truth dataset. Furthermore,

few studies analyze customer needs over multiple product categories (e.g. toothpaste, cereal, beer, etc.). Having a multi-category ground truth dataset for detecting customer needs on the keyphrase level would open the door to carrying out many different tasks, such as training a single model that detects customer needs from a range of different product categories.

To address these limitations, we outline an approach to predicting future customer needs from UGC using supervised ML. We do this by framing the problem of extracting customer needs from Reddit as a binary keyphrase classification problem where candidate keyphrases are classified at each Fixed Time Window. 15 individual corpora each representing product categories were collected by only considering posts that contain the presence of defined keyphrase(s) likely to discuss the category of analysis e.g. the defined keyphrases "'cookie" and "biscuit" make up the Cookie category. The posts from each of the categories were then preprocessed and candidate keyphrases from them were selected for the classification task. We then generated 1263 features for each of the candidate keyphrases in each product category— each coming from 10 families of features e.g. frequency-based, product-based, sentiment-based, user-based, etc. Each feature is in the form of a univariate time series, therefore associating each candidate keyphrase instance with a multivariate time series data type. We then described the process of adding the ground truth label to each candidate keyphrase instance across each of the 15 product categories. To do this we utilized the TCN- a dataset of trending keyphrase needs occurring in products each month from 2011–2021 which spans multiple product categories in the area of CPG. We made use of the dataset to indicate whether a candidate keyphrase will appear as a top customer need to be addressed in real products 1–3 years in the future. We are the first study to use TCN, without it supervised ML would not be able to be performed for the task of classifying future keyphrase customer needs. Finally, we detailed the MTSC algorithm in our study (i.e. Mini Rocket followed by Linear Ridge Regression) used to learn the relationship between the candidate keyphrases and the binary output label from TCN. To evaluate our approach 15 product categories were analyzed. In the main evaluation investigation, we showed that our approach could detect future customer needs significantly better than previous approaches and that our MTL model could detect future customer needs in categories it hadn't seen during training with performance similar to categories it had seen during training. In a further examination of our model, we showed that it could detect customer needs with lead times up to 2–3 years in advance of them occurring in products and be improved by large margins by changing the validation procedure.

The contributions of this research are as follows:

- Task of predicting future customer need keyphrases is forested with higher performance than previous approaches, thus improving the detection of unmet needs.

- Supervised ML was employed for the keyphrase classification task—made possible by the TCN dataset.

- MTL was employed by incorporating data from multiple product categories to build a single model.

- Conducted over multiple product categories on the category level (e.g. cheese) rather than the product model level (e.g. Charleville)—where other studies have tended to analyze on one product at the product level (e.g. Toyota rather than cars).

In light of these contributions, there were also various limitations, indicating areas of future work. Although our experiments were run on powerful machines we lacked a degree of resources in order to perform some highly intensive tasks. As a workaround, we included the use of undersampling in the training process and didn't employ the validation of important

algorithm hyper-parameters e.g. the *num_kernels* parameter to the MiniRocket algorithm or the *alpha* parameter to the Linear Ridge Regression algorithm (as discussed in Section 4.1). An obvious area of future work would thus be to explore more ways to reduce computational demands to run the experiment on all the training data and optimize additional hyper-parameters, although other techniques could also be used to efficiently perform hyper-parameter optimization for large datasets [149]. After generating features for the task, we didn't perform feature selection. Hence, we didn't explore which features most impacted the model or could be excluded to allow for potential increases in model performance or improvements in training times i.e. by reducing the input space. Such benefits may have allowed other important limitations of this study to be addressed e.g. the validation of model hyper-parameters due to the decrease in the initial feature input space. Recently, there has been an increase in the number of algorithms performing feature selection for multivariate time series data [150–153], so there's no reason this can't be performed in future studies. This study also has the limitation of not providing an Explainable Artificial Intelligence (XAI) analysis of the classification task, which has been a growing area of research for studies using ML on social media [154]. This could be useful for providing feature-level explanations for why a particular attribute is important for the task i.e. feature importance. This would also allow for benefits including answering questions on why a particular feature (e.g. admiration sentiment) or feature family (e.g. sentiment) is important e.g. the feature family "sentiment" does provide a crucial role in the prediction of future customer needs. This could also help provide instance-level explanations. This would allow for analysts using the prediction model to understand why a particular customer need is being predicted e.g. "vegan" for Dog Food products is predicted to be popular in products in the future due to its rising frequency, high sentiment and diverse user base discussing it. As with feature selection, there has been an increase in the number of algorithms allowing for the explainability of MTSC tasks [155–158], so there's no reason this analysis can't be performed in future studies. Similar to the limitation of not performing an XAI analysis of our model, an examination into what product categories most impacted the performance of the MTL model built in this study could be carried out. Many findings could arise from performing an analysis of this kind. These may include discovering that only a small number of product categories produce a similar performing model (in comparison to using all available categories) or finding out that some categories negatively impact the performance of the model.

Another limitation of our study is that although it is performed on multiple product categories, these categories are all in the area of CPG. It could be the case that our model only works in this area. A further analysis into this would need to be performed to test whether this is the case, requiring another ground truth dataset (which would be a very expensive task). Finally, although careful consideration is taken in the treatment of not allowing any bias in our experimental set-up (e.g. correctly splitting our data into training and testing sets), this study has the limitation in the fact that customer needs are being predicted retrospectively. This has been the stated limitation of the results from a previous study predicting future customer needs retrospectively [35] and has been the criticism of predicting election results after the fact [159]. That being said, our analysis is of interest nonetheless as we were able to map customer needs on Reddit to future needs in a dataset of real products i.e. TCN.

## Supporting information

**S1 Appendix.**
(PDF)

**S2 Appendix.**
(PDF)

**S3 Appendix.**
(PDF)

**S4 Appendix.**
(PDF)

**S5 Appendix.**
(PDF)

**S6 Appendix.**
(PDF)

**S7 Appendix.**
(PDF)

**S8 Appendix.**
(PDF)

**S9 Appendix.**
(PDF)

**S10 Appendix.**
(PDF)

**S11 Appendix.**
(PDF)

**S12 Appendix.**
(PDF)

**S13 Appendix.**
(PDF)

**S14 Appendix.**
(PDF)

**S15 Appendix.**
(PDF)

**S16 Appendix.**
(PDF)

**S17 Appendix.**
(PDF)

**S18 Appendix.**
(PDF)

## Author Contributions

**Conceptualization:** David Kilroy, Graham Healy, Simon Caton.

**Data curation:** David Kilroy.

**Formal analysis:** David Kilroy.

**Funding acquisition:** David Kilroy, Graham Healy, Simon Caton.

**Investigation:** David Kilroy, Graham Healy, Simon Caton.

**Methodology:** David Kilroy, Graham Healy, Simon Caton.

**Project administration:** David Kilroy, Graham Healy, Simon Caton.

**Resources:** David Kilroy, Graham Healy, Simon Caton.

**Software:** David Kilroy.

**Supervision:** Graham Healy, Simon Caton.

**Validation:** David Kilroy, Graham Healy, Simon Caton.

**Visualization:** David Kilroy.

**Writing – original draft:** David Kilroy, Graham Healy, Simon Caton.

**Writing – review & editing:** David Kilroy, Graham Healy, Simon Caton.

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
