## [Decision Letter · Decision Letter 0]

25 Sep 2023

PONE-D-23-20784Prediction of Future Customer Needs Using Machine Learning Across Multiple Product CategoriesPLOS ONE

Dear Dr. Kilroy,

Thank you for submitting your manuscript to PLOS ONE. After careful consideration, we feel that it has merit but does not fully meet PLOS ONE’s publication criteria as it currently stands. Therefore, we invite you to submit a revised version of the manuscript that addresses the points raised during the review process.

We look forward to receiving your revised manuscript.

Kind regards,

Toqir Rana

Academic Editor

PLOS ONE

Journal Requirements:

2. In your Methods section, please include additional information about your dataset and ensure that you have included a statement specifying whether the collection and analysis method complied with the terms and conditions for the source of the data.

This publication has emanated from research supported in part by a grant from Science 1599

Foundation Ireland under Grant number 18/CRT/6183. For the purpose of Open 1600

Access, the author has applied a CC BY public copyright licence to any Author 1601

Accepted Manuscript version arising from this submission.

DK, 18/CRT/6183, Science Foundation Ireland, https://www.sfi.ie/, No

"No"

6. Please note that in order to use the direct billing option the corresponding author must be affiliated with the chosen institute. Please either amend your manuscript to change the affiliation or corresponding author, or email us at plosone@plos.org with a request to remove this option.

8.Please include captions for your Supporting Information files at the end of your manuscript, and update any in-text citations to match accordingly. Please see our Supporting Information guidelines for more information: http://journals.plos.org/plosone/s/supporting-information. 

Additional Editor Comments (if provided):

Dear authors,

The manuscript lacks a comparative analysis with SOTA techniques, concern raised by both of the reviewers. Moreover, the article is extra ordinary lengthy. This is recommended to address these issues.

Reviewers' comments:

Reviewer's Responses to Questions

**Comments to the Author**

1. Is the manuscript technically sound, and do the data support the conclusions?

Reviewer #1: Partly

Reviewer #2: Partly

2. Has the statistical analysis been performed appropriately and rigorously? 

Reviewer #1: Yes

Reviewer #2: No

3. Have the authors made all data underlying the findings in their manuscript fully available?

Reviewer #1: No

Reviewer #2: Yes

4. Is the manuscript presented in an intelligible fashion and written in standard English?

Reviewer #1: Yes

Reviewer #2: No

5. Review Comments to the Author

Reviewer #1: The author proposed method for Prediction of Future Customer Needs Using Machine Learning Across Multiple Product Categories mainly used supervised Machine Learning (ML). The experimental analysis is not well explained, and the proposed method is worthy for investigation. The paper is lack of small issue which should be consider for improving the manuscript.

1. Author mentioned in Abstract section it is a First study while same approach already available previously.

2. Author mainly focused Dataset, features, and metrics in whole paper, so that it is missing Methods clarification properly, kindly re organized paper.

3. The author must focus on the small typos, uses of punctuation and English level throughout the manuscript.

4. Author should mention Which type of Machine Learning Algorithm used with proper explanation.

5. I will suggest author to use more exiting method for comparison of the proposed method to show the result.

6. In Conclusion part Author explained very lengthy it should be concluded in one or two paragraph, as this paper required (It should be minimized)

7. Need to modify references also, most of the references are too old.

8. Author Should add Future scope of this Article.

Reviewer #2: 1. The authors wrote “The challenging task of predicting future customer needs is performed better than previous studies” but at the same time they are claiming that no previous studies worked on predicting future customer needs. Kindly clarify it.

2. The entire manuscript needs to be scanned to address the varied language issues. For example, on line 257 “is how other approaches differ on”

3. There should be a precise name of Algorithm 1. Further, all the steps of this algorithm should be clearly elaborated.

4. The sub-algorithms of Algorithm 1 are absent in the study. For example, getDataForCategory()

These algorithms should also be written with clear and well defined instructions so that the potential reader may understand them.

5. For the evaluation, the SOTA metrics are missing. They must be used.

6. Lastly, article is too much lengthy. It must be made succinct.

6. PLOS authors have the option to publish the peer review history of their article (what does this mean?). If published, this will include your full peer review and any attached files.

Reviewer #1: **Yes: **ZULFIKAR ALI ANSARI

Reviewer #2: **Yes: **Dr Nadeem Iqbal

---

## [Author Response · Author response to Decision Letter 0]

6 Mar 2024

See attached file "Response to Reviewers.pdf"

---

## [Decision Letter · Decision Letter 1]

26 Mar 2024

PONE-D-23-20784R1Prediction of Future Customer Needs Using Machine Learning Across Multiple Product CategoriesPLOS ONE

Dear Dr. Kilroy,

Thank you for submitting your manuscript to PLOS ONE. After careful consideration, we feel that it has merit but does not fully meet PLOS ONE’s publication criteria as it currently stands. Therefore, we invite you to submit a revised version of the manuscript that addresses the points raised during the review process.

We look forward to receiving your revised manuscript.

Kind regards,

Toqir Rana, Ph.D.

Academic Editor

PLOS ONE

Reviewers' comments:

Reviewer's Responses to Questions

**Comments to the Author**

1. If the authors have adequately addressed your comments raised in a previous round of review and you feel that this manuscript is now acceptable for publication, you may indicate that here to bypass the “Comments to the Author” section, enter your conflict of interest statement in the “Confidential to Editor” section, and submit your "Accept" recommendation.

Reviewer #1: (No Response)

Reviewer #2: (No Response)

2. Is the manuscript technically sound, and do the data support the conclusions?

Reviewer #1: Yes

Reviewer #2: Partly

3. Has the statistical analysis been performed appropriately and rigorously? 

Reviewer #1: Yes

Reviewer #2: No

4. Have the authors made all data underlying the findings in their manuscript fully available?

Reviewer #1: Yes

Reviewer #2: (No Response)

5. Is the manuscript presented in an intelligible fashion and written in standard English?

Reviewer #1: Yes

Reviewer #2: Yes

6. Review Comments to the Author

Reviewer #1: Thank you to address all the comments. no need to further modifications. Proceed to Editorial board.

Reviewer #2: 1. The entire manuscript needs to be scanned to fix the typos and other grammar issues.

2. Algorithm 1 needs to be renamed. Further, normally, Algorithm and its caption are written on the top of the algorithm instructions.

3. Results of the present study must be compared with the state of the art.

4. The results must be tested with the comprehensive validation metrics.

7. PLOS authors have the option to publish the peer review history of their article (what does this mean?). If published, this will include your full peer review and any attached files.

Reviewer #1: **Yes: **ZULFIKAR ALI ANSARI

Reviewer #2: **Yes: **Dr Nadeem Iqbal

---

## [Author Response · Author response to Decision Letter 1]

2 Jun 2024

Please see attached "Response to Reviewers.pdf" file. Below is the plain text of that file. 

We thank the reviewers for their feedback and suggestions. We have attempted to address all comments (discussed below); the following details the changes made in correspondence with the reviewers and editors comments. We note that these changes are also highlighted in the submitted manuscript for convenience.

Reviewer 1

No further comments. 

Reviewer 2

1. The entire manuscript needs to be scanned to fix the typos and other grammar issues.

The manuscript has been thoroughly proofread again, changes arising from this have been made throughout (highlighted in pink).

2. Algorithm 1 needs to be renamed. Further, normally, Algorithm and its caption are written on the top of the algorithm instructions.

We have renamed Algorithm 1 to “Social Media Algorithm”. The caption for it is also now written on top of the algorithm instructions.

3. Results of the present study must be compared with the state of the art.

We have added a new section (Section 4.5) to address this. Here we compare our algorithm to available SOTA methods.

4. The results must be tested with the comprehensive validation metrics.

This is a valid point. We have added a new Appendix to address this suggestion (Appendix L). This appendix shows the precision and recall results from the sole F1 metric which is recorded in the main paper (for the binary based evaluation). This provides a more comprehensive view of performance and allows the paper to illustrate different aspects of performance that a user may value. We thank the reviewer for this suggestion and feel that it has increased both the transparency of analysis and yielded a better more rounded discussion of model performance.

---

## [Decision Letter · Decision Letter 2]

2 Jul 2024

Prediction of Future Customer Needs Using Machine Learning Across Multiple Product Categories

PONE-D-23-20784R2

Dear Dr. Kilroy,

We’re pleased to inform you that your manuscript has been judged scientifically suitable for publication and will be formally accepted for publication once it meets all outstanding technical requirements.

Kind regards,

Toqir Rana, Ph.D.

Academic Editor

PLOS ONE

Additional Editor Comments (optional):

Reviewers' comments:

Reviewer's Responses to Questions

**Comments to the Author**

1. If the authors have adequately addressed your comments raised in a previous round of review and you feel that this manuscript is now acceptable for publication, you may indicate that here to bypass the “Comments to the Author” section, enter your conflict of interest statement in the “Confidential to Editor” section, and submit your "Accept" recommendation.

Reviewer #2: (No Response)

2. Is the manuscript technically sound, and do the data support the conclusions?

Reviewer #2: Yes

3. Has the statistical analysis been performed appropriately and rigorously? 

Reviewer #2: Yes

4. Have the authors made all data underlying the findings in their manuscript fully available?

Reviewer #2: Yes

5. Is the manuscript presented in an intelligible fashion and written in standard English?

Reviewer #2: Yes

6. Review Comments to the Author

Reviewer #2: 1. Sounds better now. I recommend the publication of this article. However, grammar and language issues must be resolved.

7. PLOS authors have the option to publish the peer review history of their article (what does this mean?). If published, this will include your full peer review and any attached files.

Reviewer #2: **Yes: **Dr Nadeem Iqbal

---

## [Editor Report · Acceptance letter]

11 Jul 2024

PONE-D-23-20784R2 

PLOS ONE

Dear Dr. Kilroy, 

I'm pleased to inform you that your manuscript has been deemed suitable for publication in PLOS ONE. Congratulations! Your manuscript is now being handed over to our production team.

Kind regards, 

on behalf of

Dr. Toqir Rana 

Academic Editor

PLOS ONE